# Concentration fluctuations in growing and dividing cells: Insights into the emergence of concentration homeostasis

**Chen Jia[1], Abhyudai Singh[2], Ramon Grima[3]***

**1** Applied and Computational Mathematics Division, Beijing Computational Science Research Center, Beijing, China, **2** Department of Electrical and Computer Engineering, University of Delaware, Newark, Delaware, United States of America, **3** School of Biological Sciences, University of Edinburgh, Edinburgh, United Kingdom

* Ramon.Grima@ed.ac.uk

Concentration fluctuations in growing and dividing
cells: Insights into the emergence of concentration
homeostasis. PLoS Comput Biol 18(10):
e1010574. https://doi.org/10.1371/journal.
pcbi.1010574

Molecular Oncology, ITALY

**Data Availability Statement:** The MATLAB codes
of stochastic simulations of the full and mean-field
models can be found on GitHub via the link https://

## Abstract

Intracellular reaction rates depend on concentrations and hence their levels are often regulated. However classical models of stochastic gene expression lack a cell size description and cannot be used to predict noise in concentrations. Here, we construct a model of gene product dynamics that includes a description of cell growth, cell division, size-dependent gene expression, gene dosage compensation, and size control mechanisms that can vary with the cell cycle phase. We obtain expressions for the approximate distributions and power spectra of concentration fluctuations which lead to insight into the emergence of concentration homeostasis. We find that (i) the conditions necessary to suppress cell division-induced concentration oscillations are difficult to achieve; (ii) mRNA concentration and number distributions can have different number of modes; (iii) two-layer size control strategies such as sizer-timer or adder-timer are ideal because they maintain constant mean concentrations whilst minimising concentration noise; (iv) accurate concentration homeostasis requires a fine tuning of dosage compensation, replication timing, and size-dependent gene expression; (v) deviations from perfect concentration homeostasis show up as deviations of the concentration distribution from a gamma distribution. Some of these predictions are confirmed using data for *E. coli*, fission yeast, and budding yeast.

## Author summary

Experiments show that often the number of mRNA or protein in a cell is proportional to its volume, i.e. as a cell grows, the mRNA or protein concentration remains approximately constant. This suggests that the maintenance of a constant concentration, i.e. concentration homeostasis, is important to proper cellular function and that there are mechanisms responsible behind this regulation. However most mathematical models of gene expression do not describe the coupling of transcription and cell size; the few models that do include such a description, ignore many salient aspects of cell dynamics and hence it is presently difficult to understand how concentration homeostasis emerges. In this article,

github.com/chenjiacsrc/Concentration-homeostasis.

**Funding:** C. J. acknowledges support from National Natural Science Foundation of China with grant No. U1930402 and grant No. 12131005. A. S. is supported by the National Institute of Health Grant 1R01GM126557. R. G. acknowledges support from the Leverhulme Trust (RPG-2020-327). The funders had no role in study design, data collection and analysis, decision to publish, or preparation of the manuscript.

**Competing interests:** The authors have declared that no competing interests exist.

we develop a mathematical model of gene expression that overcomes the aforementioned difficulties; it includes a detailed description of cellular biology such as cell growth, cell division, size-dependent gene expression, DNA replication, and cell size control mechanisms that can vary with the cell cycle phase. We devise a method to approximately solve this complex model to obtain expressions for the distributions and power spectra of concentration fluctuations. This allows us to deduce that accurate concentration homeostasis results from a complex tradeoff between many mechanisms. We confirm our theoretical predictions using data for bacteria and yeast.

## Introduction

Classical models of stochastic gene expression describe the fluctuations in copy numbers of mRNAs and proteins in single cells and tissues [1–3]. These models have been successful at fitting experimental copy number distributions to estimate rate constants [4–6]. They also have provided insight into the sources of intracellular fluctuations [7, 8], in their control via feedback mechanisms [9–11] and in their exploitation to generate oscillations and multi-stable states [12, 13]. However, concentrations of mRNAs and proteins are equally and perhaps more important because cells in many cases try to maintain such a concentration—experiments have shown that for many genes in cyanobacteria [14], fission yeast [15, 16], mammalian cells [17, 18], and plant cells [19], the mRNA or protein number scales linearly with cell volume in order to maintain approximately constant concentrations, a phenomenon known as concentration homeostasis (see [20] for a recent review).

The investigation of concentration fluctuations are more complex than that of copy number fluctuations since additionally we need a description of cell volume. Commonly, the propensities of master equations are assumed to be either non volume-dependent (zero and first-order reactions) or inversely proportional to volume (second-order reaction); in the latter case the volume is fixed to some time-independent constant in simulations [21, 22]. However in growing cells, there is a periodic pattern of cell volume dynamics consisting of the increase of cell volume during the cell cycle followed by its approximate halving at cell division. In classical models of stochastic gene expression, the mRNA and protein numbers will approach a steady state, while in reality, the copy number dynamics follows a similar periodic pattern—there is an increase of copy numbers during the cell cycle followed by their approximate halving at cell division [23]. This pattern is strongly influenced by various sources of noise including the variability in cell cycle durations, the doubling of gene copy numbers upon DNA replication, and the partitioning of molecules during cell division. How noise in gene expression is regulated by different aspects of cell cycle, growth, and division have been extensively studied in the literature [24–41].

Cell volume dynamics is important in the modeling of gene expression since it strongly influences many aspects of gene expression and cell growth. To maintain concentration homeostasis, there is typically a coordination of the synthesis rate of mRNA or protein with cell volume [17]—we will refer to this mechanism as balanced biosynthesis. In addition, recent studies have also shown that cell volume will also influence the growth rate within a cell cycle, as well as the timing of gene replication and cell division in many cell types [42–45]. In particular, the amount of growth produced during the cell cycle must be controlled such that, on average, large cells at birth grow less than small ones, a strategy known as size homeostasis. To capture these phenomena, there is the need of a model of gene expression that explicitly

describes cell size dynamics; this can then be used to understand the factors giving rise to size and concentration homeostasis.

There are three popular models describing cell volume dynamics and the associated size homeostasis. The first one is the size-additive or time-additive autoregressive model [23, 46]. The time-additive model can reproduce the experimentally observed distribution of newborn sizes, which is often log-normal [46]; however, it fails to capture the fluctuations of cell cycle durations, which is usually Erlang distributed in various cell types—the doubling time distribution is left-skewed for the size-additive model and is symmetric (Gaussian) for the time-additive model, while it is right-skewed in experiments [37, 47]. The second one is the age-structured model [48–50], where the rate of cell division depends not only on cell volume, but also on the age of the cell. The third one is the multi-stage model [51, 52], where the progression of cell cycle is modeled as multiple effective cell cycle stages with the transition rate between successive stages depending on cell volume. The first two models are simpler than the third but both of them are non-Markovian. The advantages of the third model is that (i) it can reproduce the experimentally observed Erlang distribution of cell cycle durations [37, 52], (ii) it provides a convenient way to retain the Markov property which simplifies calculations, and (iii) it has a nice scaling property which naturally transforms any size control strategy into an adder [53].

Recently, a number of detailed gene expression models with both cell volume and cell cycle descriptions have been investigated [24–27]. However, most of them [24–26] did not provide an analytical theory of concentration fluctuations in growing cells. A recent study [27] made progress in this direction. In particular, a simple gene expression model in growing and dividing cells was shown to be exactly solvable when (i) the transcription activity scales linearly with cell volume and (ii) there is no variation of gene copy numbers across the cell cycle, i.e. gene replication is not taken into account. While (i) is common, it is by no means a universal phenomenon—there are several examples where there is no such scaling in both prokaryotic [54, 55] and eukaryotic cells [56–60]. The assumption behind (ii) is of course a means to simplify the model but clearly unrealistic. Relaxing any one of these two properties means that within the theoretical framework presented in [27], it is not possible to obtain an exact solution for the copy number and concentration distributions.

Here we develop a model that takes into account experimental observations of size-dependent gene expression, gene replication, gene dosage compensation, cell growth, cell division, cell cycle duration variability, and size control mechanisms that vary according to cell cycle phases. By solving the model, we obtain insight into the statistical properties of copy number and concentration fluctuations in single cells across their cell cycle, the emergence of concentration homeostasis, and the non-trivial relationship between size control mechanisms and gene expression. Some of the predictions are confirmed by analysis of published data and others provide motivation for future experiments.

## Results

### Building a computational model that captures salient features of the underlying molecular biology

We consider a biologically detailed, stochastic model of gene expression coupled to cell size dynamics with the following properties (Fig 1A). The specific meaning of all model parameters is listed in Table 1.

1) Cell volume grows exponentially with rate $g$. This assumption holds for a wide range of cell types [61–64].

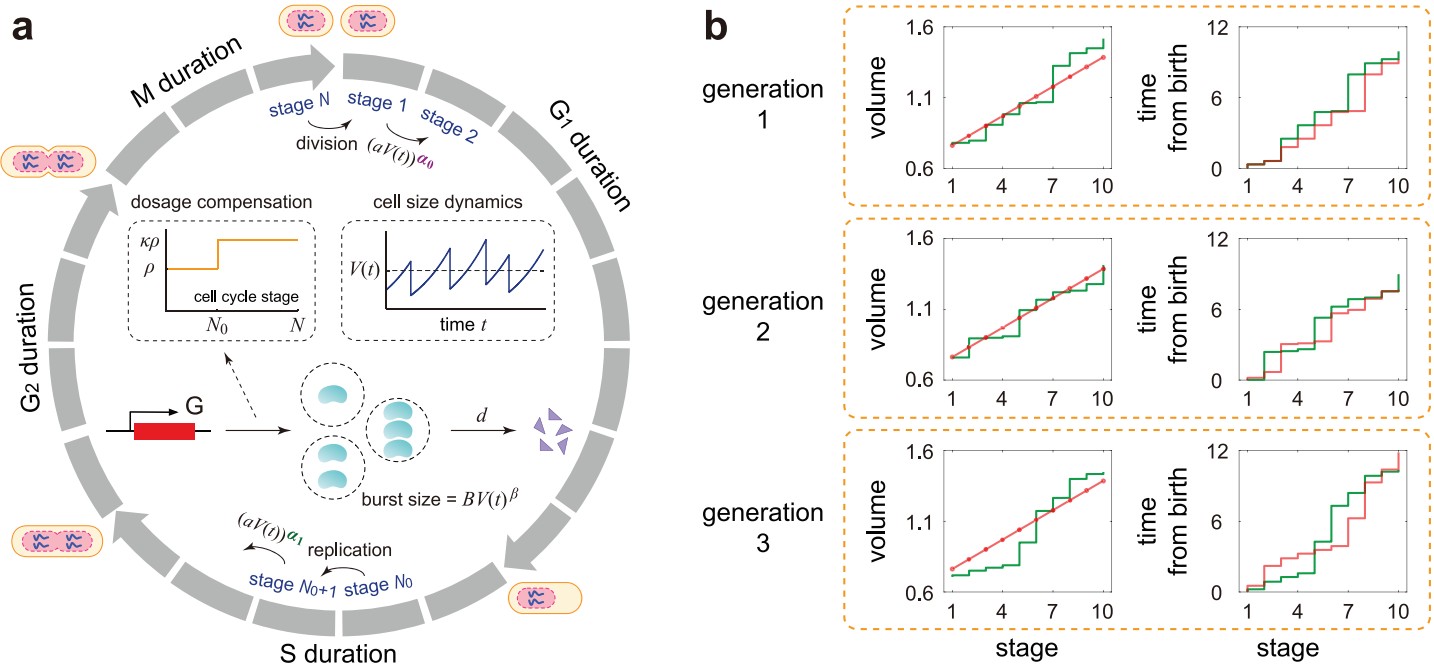

**Fig 1. Model and its mean-field approximation. A**: A detailed stochastic model with both cell size and gene expression descriptions. The model consists of $N$ effective cell cycle stages, gene replication at stage $N_0$, cell division at stage $N$, bursty production of the gene product, and degradation of the gene product. The growth of cell volume is assumed to be exponential and gene dosage compensation is induced by a change in the burst frequency at replication from $\rho$ to $\kappa\rho$ with $\kappa \in [1, 2)$ (see inset graphs). Both the transition rates between stages and the mean burst size depend on cell size in a power law form due to size homeostasis and balanced (or non-balanced) biosynthesis, respectively. The model also considers two-layer cell-size control with its strength varying from $\alpha_0$ to $\alpha_1$ upon replication. **B**: Schematics of the changes in cell volume and time from birth as a function of cell cycle stage $k$ for the first three generations in the case of $N = 10$ and $\alpha_0 = \alpha_1 = 1$. In the full model (green curve), the gene expression dynamics is coupled with the following two types of fluctuations: noise in cell volume at each stage and noise in the time interval between two stages. The reduced model (red curve) ignores the former type of fluctuations by applying a mean-field approximation while retaining the latter type of fluctuations. Note that when $\alpha_0 = \alpha_1 = 1$, it follows from Eq (3) that the mean cell volume at stage $k$ is $v_k = v_1 + (k - 1)M_0/N_0$, which linearly depends on $k$. The red dots in the stage-volume plot show the dependence of $v_k$ on $k$ and the joining of these dots by a straight red line is simply a guide to the eye. The reduced model approximates the full model excellently when $N$ is sufficiently large.

2) Before replication, the synthesis of the gene product of interest, mRNA or protein, occurs at a rate $\rho$ in bursts of a random size sampled from a geometric distribution with mean $B'$ [2] and the gene product is degraded with rate $d$ according to first-order kinetics. This can be represented by the following two effective reactions:

$$G \xrightarrow{\rho p^k(1-p)} G + kM, \quad k \geq 1, \quad M \xrightarrow{d} \varnothing, \tag{1}$$

where $G$ is the gene, $M$ is the corresponding gene product, $\rho$ is the burst frequency, and the burst size has the geometric distribution $P(k) = p^k(1 - p)$ with parameter $p = B'/(B' + 1)$. In some types of bacteria, the transcriptional activity for some promoters is constant throughout the cell cycle which also implies the transcriptional parameters $\rho$ and $B'$ are independent of cell size [54]. However, in yeast, mammalian cells, and plant cells, the products of many genes are often produced in a balanced manner: the effective synthesis rate $\rho B'$ is proportional to cell volume [15–19]. Moreover, recent studies have shown that in the presence of balanced biosynthesis, cell size controls the synthesis rate via modulation of the burst size, instead of the burst frequency [17]. To unify results in various cell types, we assume that $B' = BV(t)^\beta$, where $V(t)$ is cell volume, and $B > 0$ and $0 \leq \beta \leq 1$ are two constants. Balanced (non-balanced) biosynthesis corresponds to the case of $\beta = 1$ ($\beta = 0$).

**Table 1. Model parameters and their meaning.**

| original parameters | meaning |
|---|---|
| $g$ | growth rate of cell volume |
| $\rho$ | burst frequency before replication |
| $\kappa\rho$ | burst frequency after replication |
| $B$ | proportionality constant for the mean burst size |
| $\beta$ | degree of balanced biosynthesis |
| $d$ | degradation rate of the gene product |
| $N$ | total number of effective cell cycle stages |
| $N_0$ | number of effective cell cycle stages before replication |
| $\alpha_0$ | strength of size control before replication |
| $\alpha_1$ | strength of size control after replication |
| $a$ | proportionality constant for the transition rate between stages |
| derived parameters | meaning |
| $T \approx \log(2)/g$ | mean cell cycle duration |
| $f = 1/T$ | cell cycle frequency |
| $d_{\text{eff}} = d + (\log 2)f$ | effective decay rate of the gene product |
| $\eta = d/f$ | ratio of the degradation rate to the cell cycle frequency |
| $v_k$ | mean cell volume at stage $k$ |
| $q_k$ | transition rate from stage $k$ to the next in the mean-field model |
| $B_k$ | mean burst size at stage $k$ in the mean-field model |
| $\rho_k$ | burst frequency at stage $k$ in the mean-field model |
| $w = \log_2(v_{N_0 + 1}/v_1)$ | proportion of cell cycle duration before replication |

We emphasize that while we assume bursty expression here, our model naturally covers constitutive expression since the latter can be regarded as a limit of the former [65, 66]. In fact, when $\rho \to \infty$, $B \to 0$, and $\rho B = \bar{\rho}$ keeps constant, we have $\rho p^k(1 - p) \to \bar{\rho}V(t)^\beta$ for $k = 1$ and $\rho p^k(1 - p) \to 0$ for any $k \geq 2$. In this case, the reaction scheme given in Eq (1) reduces to

$$G \xrightarrow{\bar{\rho}V(t)^\beta} G + M, \quad M \xrightarrow{d} \varnothing,$$

which describes constitutive expression. Hence all the results below are applicable to both bursty and non-bursty expression.

3) The cell can exist in $N$ effective cell cycle stages, denoted by 1, 2, ..., $N$. Note that the $N$ effective stages do not directly correspond to the four biological cell cycle phases (G$_1$, S, G$_2$, and M). Rather, each cell cycle phase is associated with multiple effective stages (Fig 1A). The idea to model cell cycle as a series of uncoupled, memoryless stages have been successfully used to reproduce the measured variability in the durations of different cell cycle phases in various cell types [47, 67]. In addition, recent studies have revealed that the accumulation of some activator to a critical threshold may be used to promote mitotic entry and trigger cell division, a strategy known as activator accumulation mechanism [68]. For example, in fission yeast, the activator was believed to be a protein upstream of Cdk1, the central mitotic regulator, such as Cdr2, Cdc25, and Cdc13 [69–72]. Biophysically, the $N$ effective stages can be understood as different levels of the key activator that triggers cell division. Besides, another strategy called the inhibitor dilution mechanism may also be used for size control. This strategy has been observed in budding yeast [68, 73], where the decrease of the Whi5 concentration below a given threshold triggers the G$_1$/S transition allowing subsequent DNA replication and budding.

4) Gene replication occurs instantaneously when the cell transitions from a fixed stage $N_0$ to the next. Note that replication of the whole genome occurs in the S phase, which cannot be assumed to be instantaneous. However, since the replication time of a particular gene is much shorter than the total duration of the S phase, it is reasonable to consider it to be instantaneous. In addition, recent experiments have shown that the time elapsed from birth to replication for a particular gene occupies an approximately proportion of the cell cycle, which is called the stretched cell cycle model [74]. This is also consistent with our assumption that replication of the gene of interest occurs after exactly $N_0$ stages. While replication occurs after a fixed number of stages, nevertheless the time of replication is stochastic since each stage has a random lifetime (see [75] for a review about stochastic replication timing). In bacteria, there may be multiple and overlapping rounds of replication per division cycle and thus the timing of replication becomes much more complicated [76, 77]; this is not considered in the present paper.

5) Recently, there has been evidence that yeast and mammalian cells may use different size control strategies before and after replication to achieve size homeostasis [42–45]. To model this effect, we assume that the transition rate from one effective stage to the next depends on cell volume in a power law form as $(aV(t))^{\alpha_0}$ pre-replication and $(aV(t))^{\alpha_1}$ post-replication, where $a$ is a constant and $\alpha_0, \alpha_1 > 0$ are the strengths of size control before and after replication. The power law form for the rate of cell cycle progression may come from cooperativity of the key activator that triggers cell division, as explained in detail in [51, 52]. We emphasize that while this power law is compatible with certain biophysical mechanisms, it could also simply be understood as a phenomenological means to model size homeostasis, as will be explained later. Note that here we do not make any assumption about the time scale of cell cycle progression—the transition between stages can be very fast, or comparable, or very slow compared to the time scales of synthesis and degradation of the gene product.

Let $V_b$, $V_r$, and $V_d$ denote the cell volumes at birth, replication, and division, respectively. In Methods, we prove (i) the increment in the $\alpha_0$th power of cell volume before replication, $\Delta_0 = V_r^{\alpha_0} - V_b^{\alpha_0}$, has an Erlang distribution with shape parameter $N_0$ and mean $M_0 = N_0 \alpha_0 g / a^{\alpha_0}$ and (ii) the increment in the $\alpha_1$th power of cell volume after replication, $\Delta_1 = V_d^{\alpha_1} - V_r^{\alpha_1}$, is also Erlang distributed with shape parameter $N_1 = N - N_0$ and mean $M_1 = N_1 \alpha_1 g / a^{\alpha_1}$. Both $\Delta_0$ and $\Delta_1$ will be referred to generalized added volumes in what follows.

We next focus on three important special cases. When $\alpha_0 \to 0$, the transition rate from stage $k \in [1, N_0]$ to the next is a constant independent of $k$ and thus the time elapsed from birth to replication has an Erlang distribution independent of the birth volume; this corresponds to timer [78]. When $\alpha_0 = 1$, the added volume $V_r - V_b$ before replication has an Erlang distribution independent of the birth volume; this corresponds to adder. When $\alpha_0 \to \infty$, the $\alpha_0$th power of cell volume at replication, $V_r^{\alpha_0}$, has an Erlang distribution independent of the birth volume; this corresponds to sizer. In other words, $\alpha_0 \to 0, 1, \infty$ corresponds to the timer, adder, and sizer strategies before replication, respectively. Similarly, $\alpha_1 \to 0, 1, \infty$ corresponds to timer, adder, and sizer after replication, respectively. We define timer-like control to be when $0 < \alpha_i < 1$ and sizer-like control when $1 < \alpha_i < \infty$ [51].

6) After replication, the number of gene copies doubles. If the burst frequency per gene copy does not change at replication, then the total burst frequency after replication is $2\rho$. Gene dosage compensation is then modeled as a change in the burst frequency at replication from $\rho$ to $\kappa\rho$ with $1 \leq \kappa < 2$. [17, 79]. In the absence of dosage compensation, we have $\kappa = 2$. Perfect dosage compensation occurs when $\kappa = 1$; in this case, the total burst frequency for all gene copies is unaffected by replication.

7) Cell division occurs when the cell transitions from stage $N$ to stage 1. The volumes of the two daughter cells are assumed to be the same and exactly one half of the volume before division. Moreover, we assume that each gene product molecule has probability 1/2 of being allocated to each daughter cell. With this assumption, the number of gene product molecules that are allocated to each daughter cell has a binomial distribution.

We emphasize that in our model, we assume that the gene product is produced in instantaneous bursts, while in reality there is a finite time for the bursts to occur. A more general assumption is that within each cell cycle, the gene expression dynamics is characterized by the following three-stage model [2]:

$$G \xrightarrow{\rho} G^*, \quad G^* \xrightarrow{r} G, \quad G^* \xrightarrow{sV(t)^\beta} G^* + M, \quad M \xrightarrow{u} M + P,$$
$$M \xrightarrow{v} \varnothing, \quad P \xrightarrow{d} \varnothing, \tag{2}$$

where the first two reactions describe the switching of the gene between an inactive state $G$ and an active state $G^*$, the middle two reactions describe transcription and translation, and the last two reactions describe the degradation of the mRNA $M$ and the protein $P$. Here the synthesis rate of mRNA depends on cell volume via a power law form with power $\beta \in [0, 1]$. Dosage compensation can be modeled by a decrease in the gene activation rate (for each gene copy) from $\rho$ to $\kappa\rho/2$ upon replication [26, 79]. Previous studies have revealed that the bursting of mRNA and protein has different biophysical origins [80]: transcriptional bursting is due to a gene that is mostly inactive, but transcribes a large number of mRNA when it is active ($r \gg \rho$ and $s/r$ is finite), whereas translational bursting is due to rapid synthesis of protein from a single short-lived mRNA molecule ($v \gg d$ and $u/v$ is finite). Under the above timescale assumptions, both mRNA and protein are produced in a bursty manner with the reaction scheme described by Eq (1) [81–83]. The burst frequency for mRNA and protein are both $\rho$ before replication and $\kappa\rho$ after replication. The mean burst size for mRNA is $(s/r)V(t)^\beta$ and the mean burst size for protein is $(su/rv)V(t)^\beta$, both of which have a power law dependence on cell volume.

In S1 and S2 Figs, we compare the mRNA and protein distributions for the bursty model with the reaction scheme given by Eq (1) and the three-stage model with the reaction scheme given by Eq (2), where both models under consideration have a cell volume description. It can be seen that the distributions for the two models are very close to each other under the above timescale separation assumptions with the bursty model being more accurate as $r/\rho$ and $v/d$ increase. Moreover, we find that the accuracy of the bursty model is insensitive to the number of stages $N$. Here the values of $N$ are chosen so that the ratio of the average time spent in each stage ($T/N$, where $T \approx (\log 2)/g$ is the mean cell cycle duration) and the mean burst duration ($1/\rho$) ranges from $\sim 0.5 - 2$. This shows that the effectiveness of the bursty model does not require that the lifetime of a cell cycle stage is sufficient long. Due to mathematical complexity, we only focus on the bursty model in what follows—this is what we shall refer to as the full model in the rest of the paper. The consistency between the gene product distributions for the two models justifies our bursty assumption.

## Simplifying the computational model via mean-field approximation

The model described above is very complex and hence it is no surprise that the master equation describing its stochastic dynamics is analytically intractable. The difficulty in solving the statistics of gene product fluctuations stems from the fact that they are coupled to two other types of fluctuations: (i) noise in the time elapsed between two cell cycle stages; (ii) noise in cell volume when a cell cycle stage is reached. Specifically, if at time $t$ a cell enters stage $k$ and has

volume $V(t)$, then it will jump to stage $k + 1$ after a random interval $\Delta t$ whose distribution is determined by the time-dependent rate function $(aV(t))^{\alpha_i}$. Since volume growth is exponential, the volume at stage $k + 1$ is given by $V(t + \Delta t) = V(t)e^{g\Delta t}$. As can be seen from this reasoning, the fluctuations in cell volume at a certain stage depend on the fluctuations in the time to move to this stage from the previous stage.

To simplify this model, we devised a mean-field approximation on the volume dynamics, i.e. we ignore volume fluctuations at each stage but retain fluctuations in the time elapsed between two stages (Fig 1B). Specifically, we calculate an approximate mean cell size at each stage by averaging over generations. This implies that the volume change from stage $k$ to stage $k + 1$ is now deterministic and decoupled from the fluctuations in the time interval between the two stages. Within this approximation, the volume at division is twice that at birth implying that the mean cell cycle duration is $T \approx \log(2)/g$ and thus the cell cycle frequency is $f \approx g/\log(2)$. The time interval is itself exponentially distributed with rate $(av_k)^{\alpha_i}$, where $v_k$ is the mean cell size at stage $k$.

Since the exact mean cell volume at each stage is difficult to obtain, we make the approximation that the number of stages is large, which holds for most cell types and most growth conditions [37, 40, 52]. When $N \gg 1$, the fluctuations in cell volume at each stage are very small and thus can be ignored. In this case, both the generalized added volumes $\Delta_0 = M_0$ and $\Delta_1 = M_1$ can be viewed as deterministic and we have $V_d = 2V_b$. Thus the typical birth size $V_b$ is the solution of the implicit equation

$$\left(V_b^{\alpha_0} + M_0\right)^{\alpha_1/\alpha_0} + M_1 = (2V_b)^{\alpha_1}.$$

Moreover, when $N$ is large, both $v_k^{\alpha_0} - v_1^{\alpha_0}$ and $v_k^{\alpha_1} - v_{N_0+1}^{\alpha_1}$ can be viewed as deterministic and replaced by their means, as given above. Thus the typical cell size at stage $k$ is given by

$$
\begin{aligned}
v_k &= \left[v_1^{\alpha_0} + (k-1)M_0/N_0\right]^{1/\alpha_0} \quad \text{for} \quad k \in [1, N_0], \\
v_k &= \left[(v_1^{\alpha_0} + M_0)^{\alpha_1/\alpha_0} + (k - N_0 - 1)M_1/N_1\right]^{1/\alpha_1} \quad \text{for} \quad k \in [N_0 + 1, N].
\end{aligned}
\tag{3}
$$

This approximation is generally valid provided the number of cell cycle stages $N$ is large enough, which equivalently means that the variability in cell cycle duration is not very large [52]. In our model, the time from birth to replication occupies approximately a fixed proportion of the total cell cycle duration, with the proportion given by $w = \log_2(v_{N0} + 1/v_1)$. As mentioned earlier, this is consistent with the stretched cell cycle model proposed in [74]. Note that the proportion $w$ of cell cycle duration before replication is different from the proportion $N_0/N$ of cell cycle stages before replication. This is because the transition rate between cell cycle stages is an increasing function of cell size, which means that earlier (later) stages have longer (shorter) durations.

We now can construct a reduced model. Since we will be replacing a stochastic variable, the cell volume, by its mean, this is a type of mean-field approximation which has a long history of successful use in statistical physics [84]. In the reduced model, we choose the transition rate from stage $k$ to the next to be $q_k = (av_k)^{\alpha_0}$ for $k \in [1, N_0]$ and $q_k = (av_k)^{\alpha_1}$ for $k \in [N_0 + 1, N]$, the mean burst size at stage $k$ to be $B_k = Bv_k^{\beta}$, and the burst frequency at stage $k$ to be $\rho_k = \rho$ for $k \in [1, N_0]$ and $\rho_k = \kappa\rho$ for $k \in [N_0 + 1, N]$ (Fig 2); these are the same rates as in the full model but with the instantaneous volume being replaced by its mean. The microstate of the reduced model can be represented by an ordered pair $(k, n)$, where $k$ is the cell cycle stage and $n$ is the copy number of the gene product. Let $p_{k,n}$ denote the probability of observing microstate

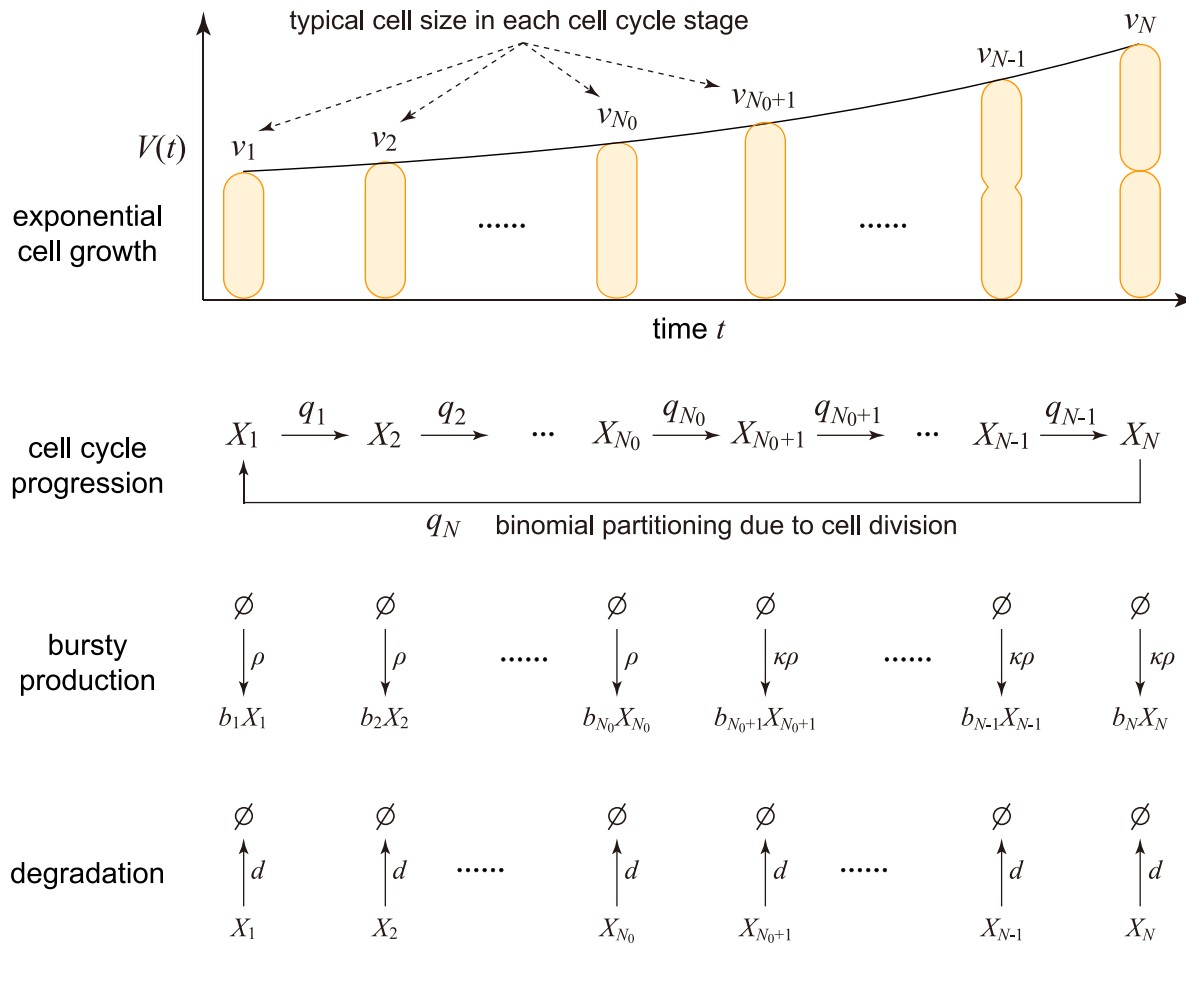

**Fig 2. Reduced model obtained by making the mean-field approximation of cell size dynamics.** Here $X_k$ denotes the gene product at stage $k$ and $b_k$ denotes the burst size at stage $k$. Moreover, $v_k$ is the typical cell size at stage $k$, $B_k$ is the mean burst size at stage $k$, and $q_k$ is the typical transition rate from stage $k$ to the next.

$(k, n)$. Then the evolution of copy number dynamics is governed by the master equation

$$
\dot{p}_{1,n} = \sum_{m=0}^{n-1} \rho_1 \chi_{1,n-m} p_{1,m} - \sum_{m=1}^{\infty} \rho_1 \chi_{1,m} p_{1,n} + (n+1) d p_{1,n+1}
$$

$$
- n d p_{1,n} + q_N \sum_{m=n}^{\infty} \binom{m}{n} \left(\frac{1}{2}\right)^m p_{N,m} - q_1 p_{1,n},
$$

(4)

$$
\dot{p}_{k,n} = \sum_{m=0}^{n-1} \rho_k \chi_{k,n-m} p_{k,m} - \sum_{m=1}^{\infty} \rho_k \chi_{k,m} p_{k,n} + (n+1) d p_{k,n+1}
$$

$$
- n d p_{k,n} + q_{k-1} p_{k-1,n} - q_k p_{k,n}, \quad 2 \leq k \leq N,
$$

where $\chi_{k,n} = [B_k/(B_k + 1)]^n[1/(B_k + 1)]$ is the burst size distribution at stage $k$, the first two terms on the right-hand side describe bursty production, the middle two describe degradation, and the last two describe cell cycle progression or binomial partitioning of gene product molecules at cell division. The master equation for the concentration dynamics can also be derived in the large molecule number limit (Section A in S1 Appendix).

Note that here the mean-field approximation is not made for the whole cell cycle, rather we make the approximation for each stage and thus different stages have different mean cell volumes. This type of piecewise mean-field approximation, as far as we know, is novel and has not been used in the study of concentration fluctuations before. In Fig 3 and S3 Fig, we make a detailed comparison between stochastic simulations of the full and mean-field models (compare dots and squares). The simulations show that they are in good agreement when $N \geq 15$

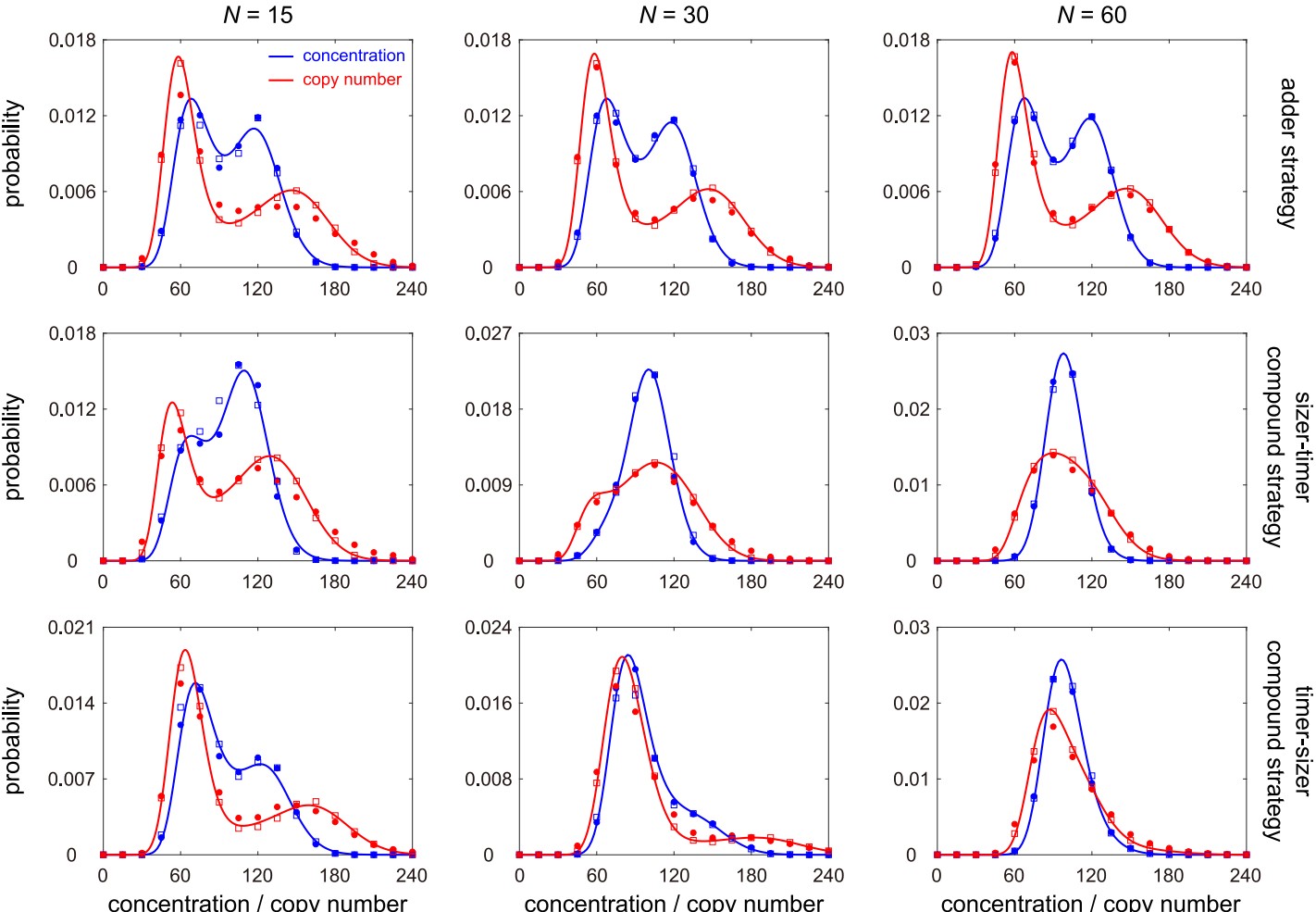

**Fig 3. Comparison between the full and mean-field models under different choices of $N$ and different size control strategies.** The blue (red) dots show the simulated concentration (copy number) distribution for the full model obtained from the stochastic simulation algorithm (SSA). The blue (red) squares show the simulated concentration (copy number) distribution for the mean-field model obtained from SSA. The blue (red) curve shows the analytical approximate concentration (copy number) distribution given in Eq (8) (Eq (15)). The full and mean-field models are in good agreement when $N \geq 15$ and they become almost indistinguishable when $N \geq 30$. Moreover, the analytical solution performs well when $N \geq 15$ and perform excellently when $N \geq 30$. The model parameters are chosen as $N_0 = 0.4N$, $B = 1$, $\beta = 0$, $\kappa = 2$, $d = 1$, $\eta = 10$, where $\eta = d/f$ is the ratio of the degradation rate to cell cycle frequency. The growth rate $g$ is determined so that $f = 0.1$. The parameters $\rho$ and $a$ are chosen so that the mean gene product number $\langle n \rangle = 100$ and the mean cell volume $\langle V \rangle = 1$. The strengths of size control are chosen as $\alpha_0 = \alpha_1 = 1$ for the upper panel, $\alpha_0 = 2$, $\alpha_1 = 0.5$ for the middle panel, and $\alpha_0 = 0.5$, $\alpha_1 = 2$ for the lower panel. When performing SSA, the maximum simulation time for each lineage is chosen to be $10^5$. To deal with the time-dependent propensities in the full model, we use the numerical algorithm described in [85, Sec. 5].

and become practically indistinguishable when $N \geq 30$. Interestingly, the condition of $N \geq 15$ holds for various cell types. Previous work estimated $N$ to be $15 - 38$ for *E. coli*, depending on temperature [52], $16 - 50$ for fission yeast, depending on temperature and culture medium [53], 59 for the cyanobacterium *S. elongatus*, 21 for rat 1 fibroblasts, 27 for human B-cells, and 50 for human mammary epithelial cells [37]. In what follows, the mean-field model, rather than the full model, will be used to study the fluctuations in gene product numbers and concentrations under different rate parameters, as well as to study the conditions for accurate concentration homeostasis.

### The concentration distribution is generally a sum of gamma distributions; for perfect concentration homeostasis, it is a single gamma

Many quantities of interest can be computed analytically using the master equation, Eq (4), of the mean-field model. Suppose that the copy number and concentration distributions of the gene product at each cell cycle stage have reached the steady state. We first compute the steady-state moment statistics. For each stage $k$, we introduce the generating function $F_k(z) = \sum_{n=0}^{\infty} p_{k,n} z^n$. Then the master equation given in Eq (4) can be converted into the following system of partial differential equations:

$$
\begin{aligned}
\partial_t F_1(z) = \quad & \rho_1 [H_1(z) - H_1(1)] F_1(z) \\
& + d(1-z)\partial_z F_1(z) + q_N F_N\left(\frac{z+1}{2}\right) - q_1 F_1(z),
\end{aligned}
$$

$$
\begin{aligned}
\partial_t F_k(z) = \quad & \rho_k [H_k(z) - H_k(1)] F_k(z) \\
& + d(1-z)\partial_z F_k(z) + q_{k-1} F_{k-1}(z) - q_k F_k(z), \quad 2 \leq k \leq N.
\end{aligned}
\tag{5}
$$

where $H_k(z) = \sum_{n=1}^{\infty} \chi_{k,n} z^n = (B_k z)/(B_k+1)[B_k(1-z)+1]$ is the generating function of the typical burst size distribution $\chi_k = (\chi_{k,n})$ at stage $k$. Let $m_{lk} = \sum_{n=0}^{\infty} n(n-1)\cdots(n-l+1)p_{k,n} = F_k^{(l)}(1)$ denote the unnormalized $l$th factorial moment of copy numbers when the cell is at stage $k$. In particular, $m_{0k} = \sum_{n=0}^{\infty} p_{k,n}$ is the probability that the system is at stage $k$. For convenience, let $m_l = (m_{lk})$ be the row vector composed of all $l$th factorial moments. Taking the $k$th derivative on both sides of Eq (5), we obtain

$$
\dot{m}_l = m_l W_{ll} + \sum_{j=0}^{l-1} m_j W_{jl},
\tag{6}
$$

where $W_{ll}$ and $W_{jl}$ are matrices defined as

$$
W_{ll} = \begin{pmatrix}
-q_1 - ld & q_1 & & & \\
& -q_2 - ld & q_2 & & \\
& & \ddots & \ddots & \\
& & & -q_{N-1} - ld & q_{N-1} \\
q_N/2^l & & & & -q_N - ld
\end{pmatrix},
\tag{7}
$$

and $W_{jl} = (l!/j!)ST^{l-j}$ with $S = \mathrm{diag}(\rho_1, \rho_2, \cdots, \rho_N)$ and $T = \mathrm{diag}(B_1, B_2, \cdots, B_N)$ being two

diagonal matrices. In steady state, we have $m_0 W_{00} = 0$ and its solution is given by $m_{0k} = 1/[\sum_{i=1}^{N} q_k/q_i]$. From Eq (6), the higher-order factorial moments of gene product numbers can be computed recursively as

$$m_l = -\sum_{j=0}^{l-1} m_j W_{jl} W_{ll}^{-1}, \quad l \geq 1.$$

In particular, the first two moments are given by $m_1 = -m_0 STW_{11}^{-1}$ and $m_2 = -2(m_1 ST + m_0 ST^2)W_{22}^{-1}$. It then follows that the mean of concentration fluctuations at stage $k$ is given by $\mu_k = m_{1k}/(m_{0k}v_k)$ and the variance at stage $k$ is given by $\sigma_k^2 = (m_{1k} + m_{2k})/(m_{0k}v_k^2) - \mu_k^2$.

We then focus on the steady-state gene product distribution measured over a cell lineage or from a population snapshot. For lineage measurements, a single cell is tracked at a series of time points, and at division one of the two daughters is randomly chosen such that we have information of the stochastic dynamics along a lineage. Once $\mu_k$ and $\sigma_k^2$ are known, the steady-state distribution of concentrations can be computed approximately as (Methods)

$$p(x) \approx \sum_{k=1}^{N} \frac{m_{0k}(\mu_k/\sigma_k^2)^{\mu_k^2/\sigma_k^2}}{\Gamma(\mu_k^2/\sigma_k^2)} x^{\mu_k^2/\sigma_k^2 - 1} e^{-\mu_k x/\sigma_k^2}, \tag{8}$$

which is a mixture of $N$ gamma distributions; the distribution of copy numbers can also be computed and is given by a mixture of $N$ negative binomial distributions (Methods). Note that the distributions given above are not exact and but serve as very good approximations. The analytical distribution agrees well with stochastic simulations when $N \geq 15$ (compare squares and curves in Fig 3 and S3 Fig). The gene product distributions can also be computed for population measurements, i.e. all single cells in a population are measured at a particular time such that we have information of the stochastic dynamics across a growing population (Methods). For the rest of this paper, we focus on lineage calculations because the results from population calculations are qualitatively the same.

We emphasize that here we do not make any assumptions about timescale separation of and cell cycle progression and gene expression dynamics. If the lifetime of the gene product is much shorter compared to the lifetime of each cell cycle stage, then the gene expression dynamics will rapidly relax to a quasi-steady state for each stage [86]. In this case, it is well known that the gene product fluctuations at each stage can be characterized by a gamma distribution in terms of concentrations [87] and by a negative binomial distribution in terms of copy numbers [88], and hence the distribution of concentrations (copy numbers) for a population of cells is naturally a mixture of $N$ gamma (negative binomial) distributions. However, the powerfulness of Eq (8)) is that it serves an accurate approximation when $N \gg 1$ without making any timescale assumptions. The effectiveness of our analytical distributions is validated in S3 Fig for three different cases: (i) the degradation rate $d$ of the gene product is much smaller than the cell cycle frequency $f$ (most proteins in bacteria and yeast); (ii) $d$ and $f$ are comparable (many mRNAs and proteins in mammalian cells); (iii) $d$ is much larger than $f$ (most mRNAs in bacteria and yeast). Please refer to Table 2 of [40] for the typical values of $d$ and $f$ in various cell types. It can be seen that our analytical distribution works reasonably well in all cases when $N \gg 1$.

It can be shown that in the special case of balanced biosynthesis ($\beta = 1$, i.e. the mean burst size scales linearly with cell size) and perfect dosage compensation ($\kappa = 1$, i.e. the total burst frequency is independent of the number of gene copies), both the mean $\mu_k$ and variance $\sigma_k^2$ of concentration fluctuations are independent of stage $k$, i.e. invariant across the cell cycle,

provided the number of gene product molecules is large (Section A in S1 Appendix). We will call this condition perfect concentration homeostasis. In this case, we have $\mu_k = \rho B/d_{\text{eff}}$ and $\sigma_k^2 = \rho B^2/d_{\text{eff}}$, where $d_{\text{eff}} = d + \log(2)f$ is the effective decay rate of the gene product due to degradation and dilution at cell division. As well, in this case, Eq (8)) reduces to a gamma distribution and the theory agrees with the classical model of Friedman et al. [87] which has no explicit cell size description.

The condition for perfect concentration homeostasis implies that the mean concentration is constant across the cell cycle, which guarantees a linear relationship between *mean* molecule number and cell volume. Note that this coincides with the definition of concentration homeostasis given in [25], while it is weaker than what is experimentally referred to as concentration homeostasis which is indicated by an approximate linear relationship between molecule number and cell volume, i.e. a high $R^2$ value for the scatter plot of molecule number versus cell volume [16, 18]. In this paper, we stick to the former definition, i.e. the linear relationship is understood in the average sense. We emphasize that perfect homeostasis ($\beta = \kappa = 1$) does not generally exist in nature, e.g. the burst frequency per gene copy does not precisely halve upon replication and hence gene dosage compensation is never perfect [17, 79]. This means that generally some deviation of the concentration distribution from gamma is expected, and this is captured by Eq (8)).

## Accurate concentration homeostasis requires a fine tuning of dosage compensation and balanced biosynthesis

Before studying concentration homeostasis in more depth, we first distinguish between two different types of concentration fluctuations. In our model, the mean $\langle c \rangle$ and variance $\sigma^2$ of concentration fluctuations (averaged over all cell cycle stages) can be computed explicitly as $\langle c \rangle = \sum_{k=1}^{N} \mu_k m_{0k}$ and $\sigma^2 = \sum_{k=1}^{N} (\sigma_k^2 + \mu_k^2) m_{0k} - \langle c \rangle^2$, where $\mu_k = m_{1k}/(m_{0k}v_k)$ and $\sigma_k^2 = (m_{1k} + m_{2k})/(m_{0k}v_k^2) - \mu_k^2$ are the mean and variance of concentrations at stage $k$, respectively, and $m_{0k}$ is the probability of the cell being at stage $k$. Then noise in concentrations can be characterized by $\phi = \sigma^2/\langle c \rangle^2$, the coefficient of variation (CV) squared of concentration fluctuations. Interestingly, we find that the total noise $\phi$ can be decomposed as $\phi = \phi_{\text{ext}} + \phi_{\text{int}}$, where

$$\phi_{\text{ext}} = \frac{\sum_{k=1}^{N} \mu_k^2 m_{0k} - \left(\sum_{k=1}^{N} \mu_k m_{0k}\right)^2}{\left(\sum_{k=1}^{N} \mu_k m_{0k}\right)^2}$$

is the extrinsic noise which characterizes the fluctuations between different stages due to cell cycle effects, and

$$\phi_{\text{int}} = \frac{\sum_{k=1}^{N} \sigma_k^2 m_{0k}}{\left(\sum_{k=1}^{N} \mu_k m_{0k}\right)^2}$$

is the intrinsic noise which characterizes the fluctuations within each stage due to stochastic bursty synthesis and degradation of the gene product.

We remind the readers that our definition of concentration homeostasis implies constancy of the mean concentration across the cell cycle [25]. Note that the extrinsic noise $\phi_{\text{ext}}$ is CV squared of the mean concentrations $\mu_k$ across all stages. Given a certain level $\phi$ of total concentration variability, the smaller the value of $\phi_{\text{ext}}$, the stronger the homeostatic mechanism. In keeping with the above definition, we define the accuracy of homeostasis as $\gamma = \phi_{\text{ext}}/\phi$, which characterizes the relative concentration of cell cycle effects in the total concentration fluctuations. Under the metric $\gamma$, accurate homeostasis occurs when the variability of the mean

concentration across the cell cycle is sufficiently small so that the total concentration variability is not dominated by it. Here we normalize $\phi_{ext}$ by $\phi$ because the cell-cycle dependence of the mean concentration will be very visible if gene expression is weakly noisy ($\phi$ is small) and will become almost invisible if gene expression is highly noisy ($\phi$ is large). Hence the visibility of cell-cycle dependence of the mean concentration should be compared to the total concentration variability $\phi$. Note that $\gamma = 0$ implies perfect homeostasis (Fig 4A and Section B in S1 Appendix). A small $\gamma$ implies an approximate linear relationship between mean copy number and cell volume, while a large $\gamma$ implies a strong nonlinear relationship between the two (see the second row in Fig 4A–4C).

We next use the metric $\gamma$ to investigate the conditions for accurate concentration homeostasis. In Fig 4D, we illustrate $\gamma$ as a function of $\beta$ and $\kappa$ when replication occurs halfway through the cell cycle ($w = 0.5$). Of particular interest is that there is a region of parameter space (shown in blue) where $\gamma$ is minimised; this is approximately given by the curve $\kappa = \sqrt{2}^{1-\beta}$ (shown by the yellow dashed line). This implies that to maintain accurate homeostasis, a lack of balanced biosynthesis requires also a lower degree of dosage compensation. The functional form of the yellow dashed line can be obtained intuitively as follows. If the gene product is very unstable, then each time that the volume changes, the concentration distribution instantaneously equilibrates and thus the mean concentration for a cell of age $t$ is given by $\rho B'/dV(t)$ before replication and is given by $\kappa \rho B'/dV(t)$ after replication. To maintain accurate homeostasis, the time-average of mean concentrations before and after replication should be equal, which shows that

$$\frac{1}{wT}\int_0^{wT}\frac{\rho B'}{dV(t)}\,dt = \frac{1}{(1-w)T}\int_{wT}^{T}\frac{\rho B'}{dV(t)}\,dt,$$

where $T \approx (\log 2)/g$ is the mean cell cycle duration and $w = \log_2(v_{N_0+1}/v_1)$ is the proportion of cell cycle before replication. This implies that $\kappa = (1-w)(2^{(\beta-1)w-1})/w(2^{\beta-1} - 2^{(\beta-1)w})$. Note that this reduces to $\kappa = \sqrt{2}^{1-\beta}$ when $w = 0.5$. We emphasize that while the above derivation is made for unstable gene products, the result also holds for stable gene products. A more rigorous derivation will be given later.

Now we look in detail at the case where there is no dosage compensation ($\kappa = 2$). In Fig 4E, we illustrate $\gamma$ as a function of $\eta$ and $\beta$, where $\eta = d/f$ is the ratio of the degradation rate to cell cycle frequency (a measure of the stability of the gene product). Note that $\gamma$ becomes smaller as $\eta$ decreases, implying that stable gene products (e.g. most proteins) give rise to better concentration homeostasis than unstable ones (e.g. most mRNAs). The combined results from Fig 4D and 4E indicate that homeostasis is the weakest when $\beta = 1$, $\kappa = 2$, and $\eta$ is large. This shows that balanced biosynthesis, weak dosage compensation, and large degradation rates together can break homeostasis.

Next we look in detail at the case where synthesis is non-balanced ($\beta = 0$). In Fig 4F, we illustrate $\gamma$ as a function of $\kappa$ and $w$, where $w = \log_2(v_{N_0+1}/v_1)$ is the proportion of cell cycle before replication. Interestingly, we find that $\gamma$ is remarkably small when $\kappa \approx \sqrt{2} \approx 1.4$ and $w \approx 0.5$. The optimal values of $\kappa$ and $w$ are stable and depend very less on other parameters. This shows that when $\beta = 0$, accurate homeostasis can still be obtained at an intermediate strength of dosage compensation when replication occurs halfway through the cell cycle.

## Precise determination of the onset of concentration homeostasis using the power spectrum of fluctuations

Generally the time traces of gene product numbers increase from birth up till division time, at which point they halve. This periodicity would be also expected to some extent in concentrations.

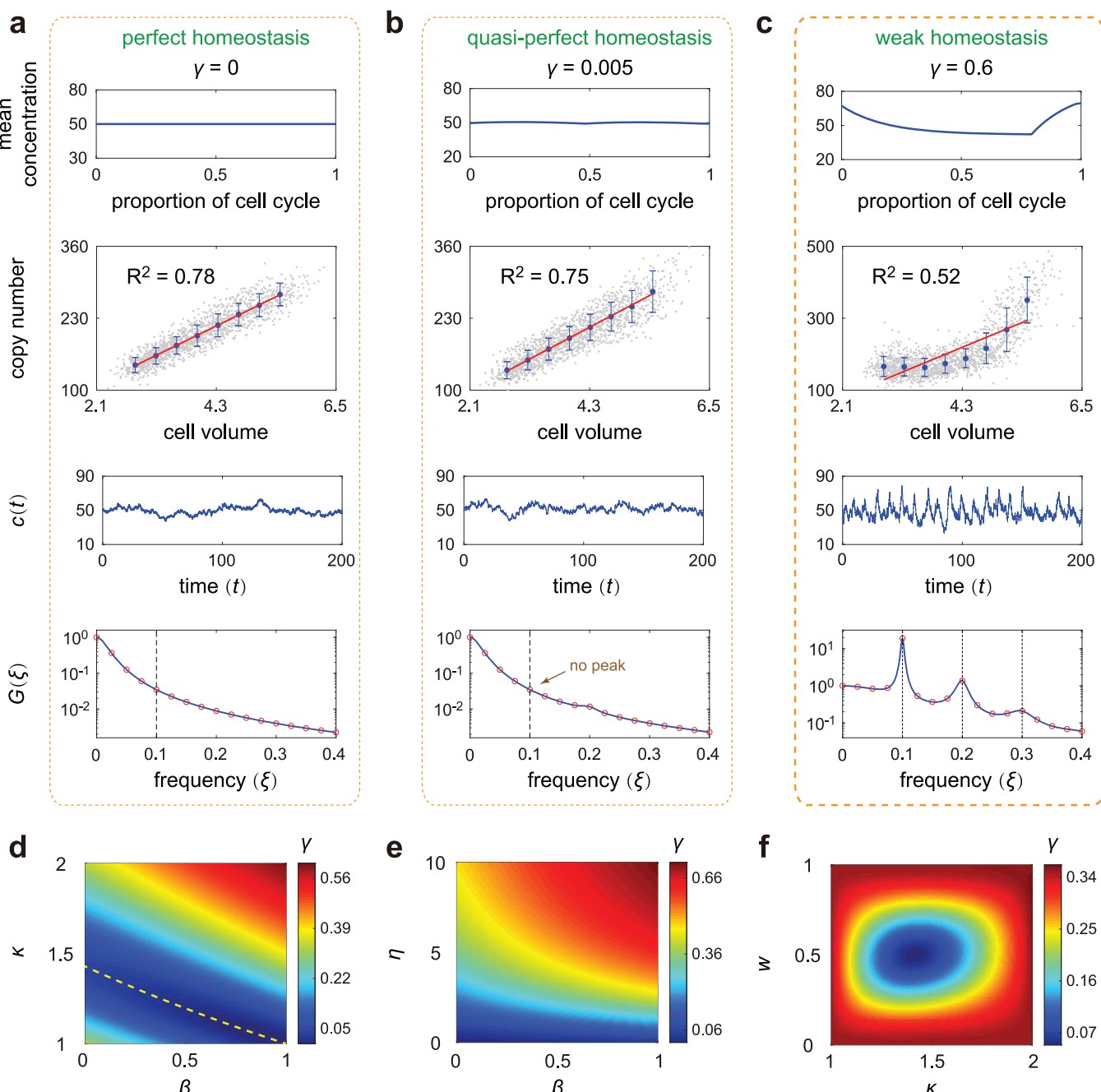

**Fig 4. Concentration homeostasis and its relation to concentration oscillations. A**: Perfect homeostasis. The first row shows the mean concentration $\mu_k$ at cell cycle stage $k$ versus the proportion of cell cycle before stage $k$, which can be computed explicitly as $w_k = \log_2(v_k/v_1)$. The second row shows the scatter plot of molecule number versus cell volume obtained from SSA. Here data are sorted into 8 bins according to cell volume, and the mean and variance of molecule numbers within each bin are shown by the blue dot and the error bar. The red curve is the regression line for the scatter plot. The third row shows typical time traces of concentrations, where $c(t)$ denotes the concentration at time $t$ for a single cell lineage. The fourth row shows the theoretical (blue curve) and simulated (red circles) power spectra of concentration fluctuations. The theoretical spectrum is computed from Eq (9), while the simulated spectrum is obtained by means of the Wiener-Khinchin theorem, which states that $G(\xi) = \lim_{T\to\infty}\langle|\hat{c}_T(\xi)|^2\rangle/T$, where $\hat{c}_T(\xi) = \int_0^T c(t)e^{-2\pi i \xi t}dt$ is the truncated Fourier transform of a single concentration trajectory over the interval $[0, T]$ and the angled brackets denote the ensemble average over trajectories. The vertical lines show the cell cycle frequency $f$ and its harmonics. **B**: Same as A but for quasi-perfect homeostasis. **C**: Same as A but for weak homeostasis. **D**: Heat plot of $\gamma$ (the accuracy of concentration homeostasis) versus $\beta$ (the degree of balanced biosynthesis) and $\kappa$ (the strength of dosage compensation) when replication occurs halfway through the cell cycle ($w = 0.5$). The yellow dashed line shows the exponential curve $\kappa = \sqrt{2}^{1-\beta}$, around which homeostasis is accurate. **E**: Heat plot of $\gamma$ versus $\beta$ and $\eta$ (the stability of the gene product) when there is no dosage compensation ($\kappa = 2$). Stable gene products give rise to more accurate homeostasis than unstable ones. **F**: Heat plot of $\gamma$ versus $\kappa$ and $w$ (the proportion of cell cycle before replication) when synthesis is non-balanced ($\beta = 0$). Homeostasis is the most accurate when $\kappa = \sqrt{2}$ and $w = 0.5$. See Section D in S1 Appendix for the technical details of this figure.

However clearly if there is strong concentration homeostasis, no such periodic behaviour would be visible. Hence it stands to reason that a lack of a peak in the power spectrum of concentration oscillations (computed from a single lineage) can be interpreted as strong concentration homeostasis. Stochastic simulations, shown in Fig 4A–4C, indicate that this intuition is indeed the case: as $\gamma$ increases (from left to right), the scatter plot of mean molecule number versus cell volume displays an increasingly nonlinear relationship with smaller $R^2$ value, the trajectory of concentrations becomes more oscillatory, and the power spectrum switches from monotonic to a peaked one at a non-zero value of frequency.

We next study the power spectrum analytically. Recall that the autocorrelation function $R(t)$ of concentration fluctuations is defined as the steady-state covariance between $c(0)$ and $c(t)$, where $c(t)$ denotes the concentration at time $t$ for a single cell lineage. Because the propensities of the reactions in our model are linear in gene product numbers, the autocorrelation function $R(t)$ can be computed in closed form as (Section C in S1 Appendix)

$$R(t) = (m_1 + m_2)V^{-1}e^{W_{11}t}V^{-1}\mathbb{1} - (m_1 V^{-1}\mathbb{1})^2$$
$$+ \int_0^t m_1 V^{-1}e^{W_{00}s}STe^{W_{11}(t-s)}V^{-1}\mathbb{1}ds, \tag{9}$$

where $V = \mathrm{diag}(\nu_1, \cdots, \nu_N)$ and $\mathbb{1} = (1, \cdots, 1)^T$. The power spectrum $G(\xi) = \int_{-\infty}^{\infty} R(|t|)e^{-2\pi i \xi t}dt$ is then the Fourier transform of the autocorrelation function which can be computed straightforwardly. Note that the power at zero frequency, $G(0)$, characterizes the strength of stochasticity that is not owing to oscillations. Throughout the paper, we normalize the power spectrum so that $G(0) = 1$. In Section C in S1 Appendix, we show that the power spectrum is a weighted sum of $N$ Lorentzian functions, which either decreases monotonically or has an off-zero peak around the cell cycle frequency $f$ (see the fourth row in Fig 4A–4C). The height of the non-zero peak (relative to the power at zero frequency) acts as a measure of the regularity of oscillations. Furthermore, when $N \gg 1$, the power spectrum also has peaks around the higher order harmonics of $f$ (Fig 4C). This is probably because an arbitrary periodic function with frequency $f$, when expanded as Fourier series, cannot be solely described by a sinusoidal function with frequency $f$, but needs components with higher order harmonics.

Interestingly, we find that the power spectrum of concentration fluctuations can be used to precisely determine the onset of concentration homeostasis. Complex computations show that the height $H$ of the off-zero peak, which characterizes the regularity of concentration oscillations, is proportional to both $N^2$ and (Section C in S1 Appendix)

$$C(\beta, \kappa, w) = |1 - \kappa 2^{\beta-1} + (\kappa - 1)2^{(\beta-1)w}e^{2\pi w i}|^2. \tag{10}$$

Generally, the condition $C(\beta, \kappa, w) = 0$ is the necessary condition for the power spectrum to lack an off-zero peak which means that strong homeostasis is present. This can be achieved when $\beta = \kappa = 1$ (balanced biosynthesis and perfect dosage compensation) which implies also $\gamma = 0$. However interestingly, it can be achieved also when balanced biosynthesis is broken. When $\beta \neq 1$, Eq (10) vanishes when $1 - \kappa 2^{\beta-1} + (\kappa - 1)2^{(\beta-1)w}\cos(2\pi w) = 0$ and $\sin(2\pi w) = 0$, i.e. $\kappa = \sqrt{2}^{1-\beta}$ and $w = 0.5$. In this case, we have $\gamma \neq 0$, which means that there is still some variation of the mean concentration across the cell cycle. As a result, we refer to this case as quasi-perfect homeostasis (Fig 4B). This also corresponds to the minimum shown by the yellow dashed line in Fig 4D. In particular, when $\beta = 0$, quasi-perfect homeostasis is obtained when $\kappa = \sqrt{2}$ and $w = 0.5$. This also corresponds to the dark blue region in Fig 4F. The optimal value of $\kappa \approx 1.4$ is comparable to the experimental values of $\kappa$ measured for *Oct4* and *Nanog* genes in mouse embryonic stem cells [79]. However generally the conditions for strong

homeostasis, i.e. no peak in the power spectrum, appear quite restrictive ($\beta = \kappa = 1$ or $\beta < 1$, $\kappa = \sqrt{2}^{1-\beta}$, and $w = 0.5$). In bacteria and budding yeast, there has been some evidence that dosage compensation is not widespread [54, 89]. In fission yeast [90] and human cells [91], examples where expression is balanced but the effects of replication are not completely buffered by dosage compensation are starting to be uncovered. In addition, replication may not occur halfway through the cell cycle (please refer to Table 2 in [40] for the typical value of $w$ in various cell types). Hence our theory predicts that strong homeostasis is probably rarely seen in nature.

It has been shown [40] that when synthesis is non-balanced ($\beta = 0$), the time traces of copy numbers for unstable gene products display the strongest oscillation regularity when $w = 0.5$; here we have shown that the time traces of concentrations display the weakest regularity when $w = 0.5$ (S4 Fig). This shows that copy number and concentration time traces may exhibit totally different dynamical behaviors.

To better understand concentration oscillations and its relation to homeostasis, we depict both $\gamma$ and $H$ as a function of $B$ and $\langle n \rangle$ (S5 Fig). As expected, the off-zero peak becomes lower as $B$ increases and as $\langle n \rangle$ decreases since both of them correspond to an increase in concentration fluctuations which counteracts the regularity of oscillations; noise above a certain threshold can even completely destroy oscillations. Furthermore, we find that $\gamma$ and $H$ have similar dependence on $B$ and $\langle n \rangle$. This again shows that the occurrence of concentration oscillations is intimately related to the visibility of cell cycle effects in concentration fluctuations.

## A sizer-timer or adder-timer size control strategy maintains concentration homeostasis while minimising concentration noise

Recently, evidence has emerged that budding yeast, fission yeast, and mammalian cells may exhibit multiple layers of size control within each cell cycle, e.g. there can be different size control strategies at work before and after replication [42–45]. A natural question is why eukaryotic cells apply such two-layer control strategies to achieve size homeostasis. In our model, we assume that the size control strength varies from $\alpha_0$ to $\alpha_1$ upon replication. In Methods, we show that when the variability in cell size is small, the birth size $V_b$ and the division size $V_d$ are interconnected by $V_d = 2^{1-\alpha} V_b + \epsilon$, where $\alpha = w\alpha_0 + (1 - w)\alpha_1$ is the effective size control strength over the whole cell cycle and $\epsilon$ is a noise term independent of $V_b$. The cell behaves as an overall timer/adder/sizer when $\alpha \rightarrow 0, 1, \infty$, respectively. Note that in real systems, $\alpha$ is never much larger than 1—for fission yeast which is known to have a sizer-like mechanism, it has recently been estimated that $\alpha$ is $1.4 - 2.1$ [53].

To understand the effect of size control on concentration homeostasis, we illustrate $\gamma$ as a function of $\beta$, $\alpha_0$, and $\alpha_1$ (Fig 5A and 5B). Recall that balanced biosynthesis, weak dosage compensation, and unstable gene products may break concentration homeostasis. When synthesis is balanced ($\beta = 1$) and dosage compensation is not very strong ($\kappa$ is not very close to 1), we find that homeostasis depends strongly on size control and its accuracy is significantly enhanced if cells use different size control strategies before and after replication (see also S6 Fig). Homeostasis is the strongest if (i) the cell behaves as a sizer before replication and a timer after replication or (ii) the cell behaves as a timer before replication and a sizer after replication. Thus such sizer-timer or timer-sizer compound control strategies may help eukaryotic cells maintain accurate homeostasis without any clearly distinguishable concentration oscillations (Fig 5E) for genes with a wide range of dynamic parameters. In contrast, when synthesis is non-balanced ($\beta = 0$) such as in bacteria, we find that homeostasis is insensitive to size control (S6 Fig).

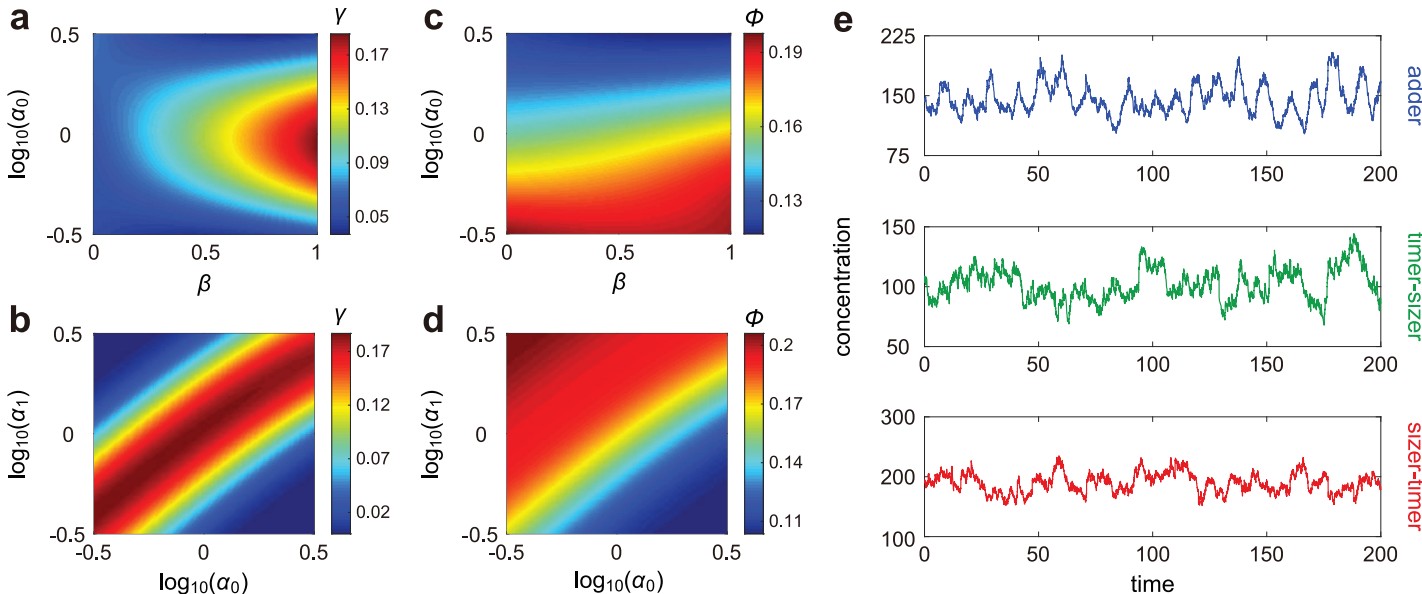

**Fig 5. Influence of size control on concentration homeostasis and concentration noise. A,B**: Heat plot of $\gamma$ (the accuracy of concentration homeostasis) versus $\beta$ (the degree of balanced biosynthesis), $\alpha_0$ (the strength of size control before replication), and $\alpha_1$ (the strength of size control after replication). When synthesis is balanced ($\beta = 1$), concentration homeostasis is accurate when the cell applies different size control strategies before and after replication. Both the timer-sizer and sizer-timer compound strategies enhance the accuracy of homeostasis. **C,D**: Heat plot of $\phi$ (noise in concentration) versus $\beta$, $\alpha_0$, and $\alpha_1$. While the timer-sizer compound strategy leads to accurate concentration homeostasis, it results in large gene expression noise. The sizer-timer compound strategy both maintains homeostasis and reduces noise. The model parameters are chosen as $N = 50$, $N_0 = 21$, $\rho = 4.7$, $B = 1$, $\kappa = 2$, $d = 0.4$, $\eta = 4$ in A-D, $\alpha_1 = 1$ in A,C, and $\beta = 1$ in B,D. The growth rate $g$ is determined so that $f = 0.1$. The parameter $a$ is chosen so that the mean cell volume $\langle V \rangle = 1$. **E**: Typical time traces of concentration under three different size control strategies: adder (upper panel), timer-sizer (middle panel), and sizer-timer (lower panel). See Section E in S1 Appendix for the technical details of this figure.

Furthermore, we find that size control also affects gene expression noise. Fig 5C and 5D illustrate the noise $\phi$ as a function of $\beta$, $\alpha_0$, and $\alpha_1$, from which we can see that concentration noise depends only weakly on $\beta$ but depends strongly on size control. Noise is minimised when the cell behaves as a sizer before replication and a timer after replication, and is maximized when the control strategy changes from timer to sizer at replication, whether synthesis is balanced or not (Fig 5D and S6 Fig). This suggests that while the timer-sizer compound strategy is conducive to concentration homeostasis, it gives rise to large concentration noise (Fig 5E). However, the sizer-timer compound strategy both maintains homeostasis and reduces noise. Note that while sizer-timer is optimal, the adder-timer compound strategy is still ideal in maintaining homeostasis and reducing noise since it also falls in the blue regions in Fig 5B and 5D.

In a recent work [45], it was found that both the duration of the $G_1$ phase and the added volume in $G_1$ decrease with birth volume in two types of cancerous epithelial cell lines (HT29-hgem and HeLa-hgem), indicating that the control strategy before replication is sizer-like. Moreover, they also found that the duration of the S-$G_2$ phase decreases and the added volume in S-$G_2$ increases with the volume at the $G_1$/S transition in both cell types, showing that the cell uses a timer-like strategy after replication. Similar results have been found in yeast [42–44]. Our results may explain why the sizer-timer compound strategy is preferred over others and hence commonly used in nature.

## Bimodality in the concentration distribution is rarer than bimodality in the copy number distribution

To gain a deeper insight into gene product fluctuations, in Fig 6A–6C we contrast the copy number and concentration distributions. Clearly, the shapes of the two can be significantly

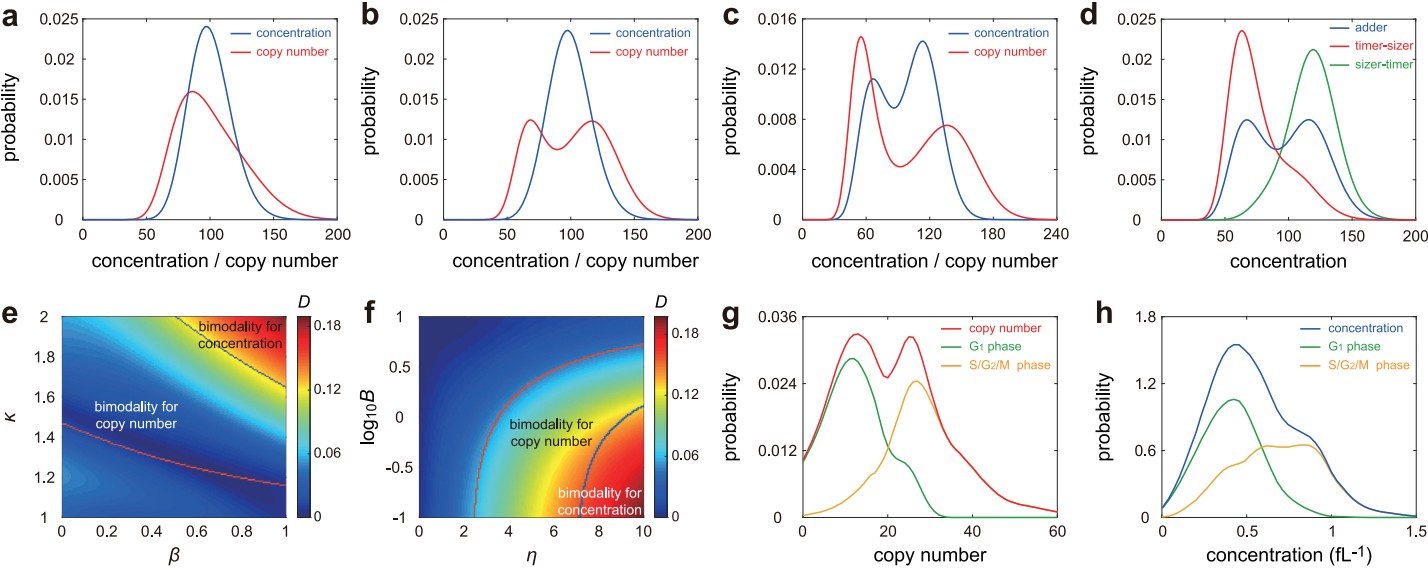

**Fig 6. Influence of model parameters on copy number and concentration distributions. A-C:** The copy number (red curve) and concentration (blue curve) distributions display different shapes under different choices of parameters. A Both distributions are unimodal. B The concentration distribution is unimodal while the copy number one is bimodal. C Both distributions are bimodal. **D:** Concentration distributions under three different size control strategies: adder (blue curve), timer-sizer (red curve), and sizer-timer (green curve). Two-layer control strategies lead to unimodal concentration distributions that are not far from gamma. **E,F:** Heat plot of the Hellinger distance $D$ of the concentration distribution from its gamma approximation versus $\beta$ (the degree of balanced biosynthesis), $\kappa$ (the strength of dosage compensation), $\eta$ (the stability of the gene product), and $B$ (the mean burst size). The violet (blue) curve encloses the region for the copy number (concentration) distribution being bimodal. The bimodal region for the copy number distribution is much larger than that for the concentration distribution. **G,H:** The mRNA copy number (red curve) and concentration (blue curve) distributions for the *HTB2* gene in budding yeast measured using smFISH. The green (yellow) curves show the distributions of cells in the $G_1$ ($G_2$-S-M) phase. No apparent dosage compensation was observed with the mean copy number (concentration) in $G_2$-S-M being 2.22 (1.66) times greater than that in $G_1$. The data shown are published in [92]. See Section F in S1 Appendix for the technical details of this figure.

different; for some parameter sets, one is unimodal and the other is bimodal. Both distributions can display bimodality, due to a change in the burst frequency upon replication; however, the concentration distribution is generally less wide than the copy number distribution, which agrees with experimental observations [18].

Recall that concentration has the gamma distribution when $\beta = \kappa = 1$ and $\langle n \rangle \gg 1$. To understand the deviation from gamma, we depict the Hellinger distance $D$ of the concentration distribution from its gamma approximation as a function of $\beta$, $\kappa$, $\eta$, and $B$ (Fig 6E and 6F). Note that Fig 6E is similar to Fig 4D, implying that the deviation from gamma is closely related to concentration homeostasis. Accurate homeostasis usually results in a concentration distribution that is gamma-shaped. In particular, stable gene products usually have a unimodal concentration distribution that is not far from gamma. Since sizer-timer and timer-sizer compound strategies enhance the accuracy of homeostasis, they both lead to a smaller deviation from gamma (Fig 6D). In Fig 6E and 6F we show the regions of parameter space where the concentration and copy number distributions are unimodal and bimodal. Both distributions display bimodality when $\beta$, $\kappa$, and $\eta$ are large and $B$ is small. However, the region of parameter space for observing bimodality in copy number distributions is much larger than that for concentration distributions. This shows that the number of modes of the concentration distribution is always less than or equal to that of the copy number distribution. The situation that the concentration distribution has more modes than the copy number distribution is not found, according to simulations of large swathes of parameter space. Bimodality in the concentration distribution occurs only in the presence of balanced biosynthesis, weak dosage compensation, large degradation rates, and small burst size.

## Confirmation of theoretical predictions using bacterial and yeast data

Next we use published experimental data to test some of the predictions of our model: (i) unstable gene products can have a concentration distribution with either the same or a smaller number of modes than the copy number distribution; (ii) strong concentration homeostasis is rare, i.e. while the gene product numbers may scale approximately linearly with cell volume, nevertheless oscillations in concentration due to the periodicity of cell division are still detectable; (iii) deviations from perfect homeostasis result in deviations of the concentration distribution from gamma. We used three data sets reporting mRNA or protein measurements for bacteria, budding yeast, and fission yeast, to test these results, as follows.

To test prediction (i), we used the population data collected in diploid budding yeast cells using single-molecule mRNA fluorescence in situ hybridization (smFISH) [92]. In this data set, the mRNA abundance and cell volume are measured simultaneously for four genes: *HTB1*, *HTB2*, *HHF1*, and *ACT1*, with the cell cycle phase ($G_1$, S, and $G_2$/M) of each cell being labelled. These mRNAs are produced in a balanced manner [92] and expected to be unstable since the median mRNA half-life in budding yeast is $\sim 20$ min [93], which is much less than the mean cell cycle duration of $1.5 - 2.5$ hr [94, 95]. Interestingly, for *HTB2*, we observed an apparent bimodal distribution for the mRNA number and a unimodal distribution for the mRNA concentration (shown by the red and blue curves in Fig 6G and 6H). Calculating the distributions conditional on the $G_1$ and S-$G_2$-M phases (shown by the green and yellow lines) clarifies that the bimodality stems from replication and a lack of dosage compensation. Specifically, since the volume distribution for the population of cells is unimodal (S7 Fig), the bimodal copy number distribution must come from replication and weak dosage compensation, rather than from a bimodal volume distribution and strong dosage compensation. Similar distribution shapes were also observed for *HTB1* and *ACT1*; for *HHF1*, both distributions are unimodal (S8 Fig).

To test predictions (ii) and (iii), we used the lineage data collected in *E. coli* and haploid fission yeast cells using a mother machine [23, 96]. In the *E. coli* data set [23], the time course data of cell size and fluorescence intensity of a stable fluorescent protein were recorded for 279 cells lineages under three growth conditions. In the fission yeast data set [96], similar data for a stable fluorescent protein were monitored for 10500 cell lineages under seven growth conditions. For both cell types, we illustrate the change in the mean concentration across the cell cycle, as well as the distribution and power spectrum of concentration fluctuations, in Fig 7A–7C and 7E–7G. In agreement with prediction (ii), from Fig 7A and 7B we see that for *E. coli*, while the mean concentration is practically invariant across the cell cycle, nevertheless homeostasis is not strong enough to suppress concentration oscillations whose signature is a peak at a non-zero frequency in the power spectrum. A similar phenomenon was also observed for fission yeast (Fig 7E and 7F). The higher-order harmonics of concentration fluctuations (second and third peaks in the spectrum) can be seen for fission yeast but not for *E. coli*; this is likely due to a stronger periodicity (smaller variability) in cell cycle duration in fission yeast compared to that in *E. coli* [52, 53]. Finally, in agreement with prediction (iii), from Fig 7D and 7H we see that for both organisms, the Hellinger distance $D$, which measures the distance of the concentration distribution from a gamma, is positively correlated with the non-dimensional parameter $\gamma$ and the off-zero peak height $H$, both of which measure how strong concentration homeostasis is.

## Discussion

In this paper, we have developed and solved a complex model of gene product fluctuations which presents a unified description of intracellular processes controlling cell volume, cell

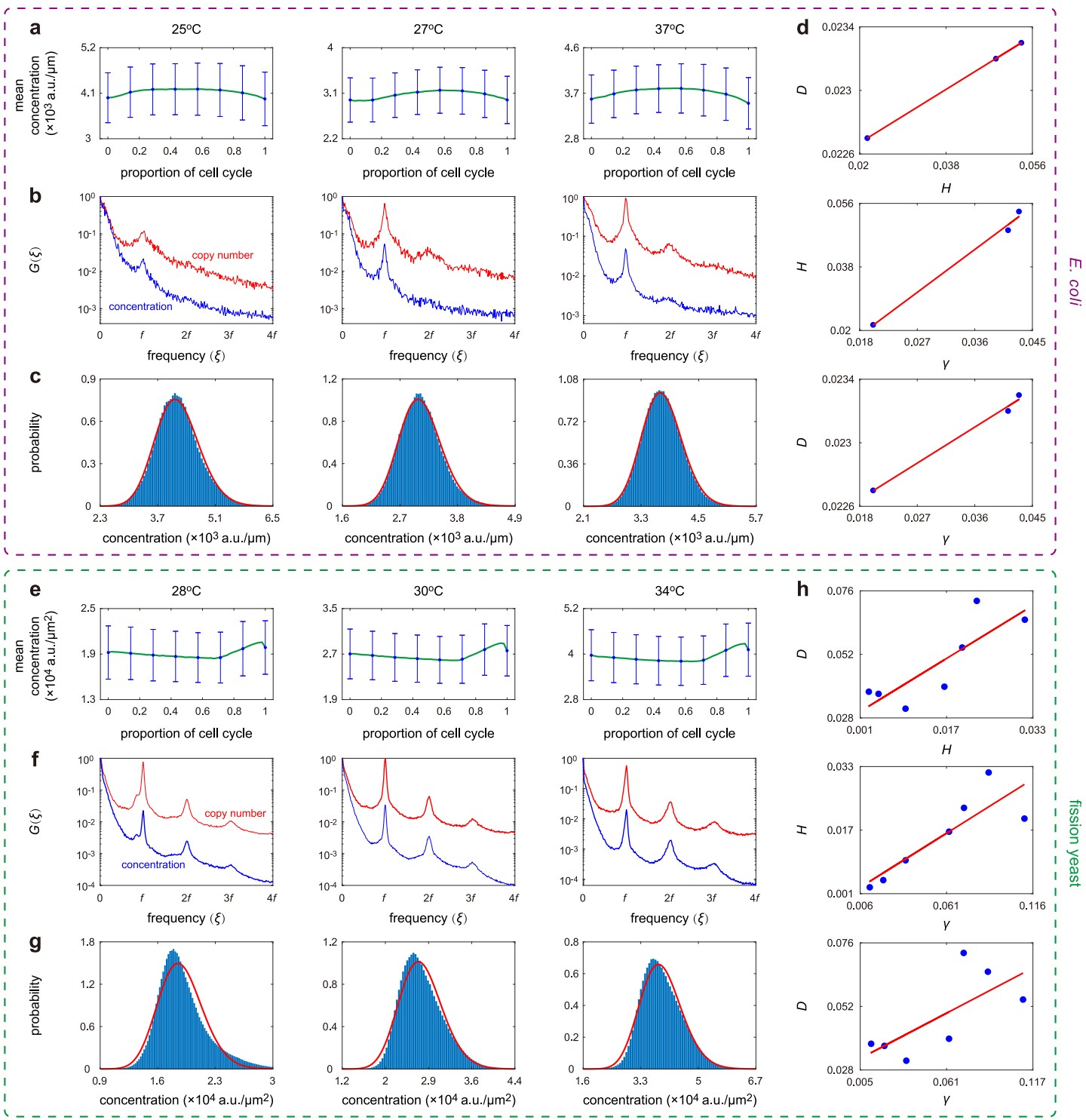

**Fig 7. Analysis of single-cell lineage data of concentration fluctuations in _E. coli_ and fission yeast.** The data shown are published in [23, 96]. The _E. coli_ data set contains the lineage measurements of both cell length and fluorescence intensity of a stable fluorescent protein at three different temperatures (25°C, 27°C, and 37°C). The fission yeast data set contains the lineage measurements of both cell area and fluorescence intensity of a stable fluorescent protein under seven different growth conditions (Edinburgh minimal medium (EMM) at 28°C, 30°C, 32°C, and 34°C and yeast extract medium (YE) at 28°C, 30°C, and 34°C). **A**: The change in the mean concentration across the cell cycle for _E. coli_ under three growth conditions. Here, we divide each cell cycle into 50 pieces and compute the mean and standard deviation (see error bars) of all data points that fall into each piece. The CV squared of the mean concentrations of all pieces gives an estimate of $\phi_{\text{ext}}$ and the CV squared of the concentrations of all cells gives an estimate of $\phi$, from which $\gamma$ can be inferred. **B**: Power spectra of copy number (red curve) and concentration (blue curve) fluctuations under three growth conditions. **C**: Experimental concentration distributions (blue bars) and their gamma approximations (red curve) under three growth conditions. **D**:

Scatter plots between $\gamma$ (the accuracy of concentration homeostasis), $H$ (the height of the off-zero peak of the power spectrum), and $D$ (the Hellinger distance between the concentration distribution from its gamma approximation) under three growth conditions. **E-H**: Same as A-D but for fission yeast. Here E-G only show the three growth conditions for YE, while H shows the scatter plots between $\gamma$, $H$, and $D$ under all seven growth conditions (for both EMM and YE). Both cell types show remarkable positive correlations between $\gamma$, $H$, and $D$.

cycle phase, and their coupling to gene expression. In our model, cell cycle and cell volume act on gene expression mainly by (i) the dependence of the burst size on cell volume; (ii) the increase in the burst frequency upon replication; (iii) the change in size control strategy upon replication; (iv) the partitioning of molecules at division. Point (iv) strongly affects copy number fluctuations, while it has little influence on concentration fluctuations. In our model, there is a strong coupling between gene expression dynamics, cell size dynamics, and cell cycle events, mainly due to size-dependent synthesis and cell size control. The analytical difficulties in solving a model with considerable biological details were circumvented by means of a novel type of mean-field approximation which involves neglecting cell size fluctuations such that the cell size at a given fraction of the cell cycle is independent of the generation number. This approximation disentangled the coupling between gene expression and cell volume and led to a reduced model from which we derived expressions for the moments, distributions, and power spectra of both gene product number and concentration fluctuations. We showed by simulations that the full and reduced models are in good agreement over all of parameter space provided the number of cell cycle stages $N$ is greater than 15 (which experimental data shows to be the case for most cell types).

We note that whilst the majority of gene expression models have no cell size description, recently a few pioneering papers have considered extensions to include such a description. In [25], it was shown that concentration homeostasis can arise when RNA polymerases are limiting for transcription and ribosomes are limiting for translation. This model takes into account gene replication and cell division but neglects dosage compensation and multi-layer size control, and some of the mechanistic details are specific to *E. coli*. In addition, the analysis is deterministic and while stochastic simulations are presented, a detailed study of fluctuations in molecule number and concentration is missing. In [26], mRNA number distributions are approximated by a negative binomial using a model that takes into account replication, division, dosage compensation, and balanced biosynthesis. The main limitations of this approach is its lack of flexibility to capture complex distributions and the neglection of fluctuations in cell volume dynamics. In [27], a simplified model is considered whereby replication and multi-layer size control are neglected and dosage compensation is assumed to be perfect. The main limitations of this approach is the assumption of stochastic concentration homeostasis (SCH) [27], which will be broken when the synthesis of the gene product is not balanced or when dosage compensation is not perfect. Specifically, when synthesis is balanced and when dosage compensation is perfect, our model reduces to one of the models studied in [27]; this model satisfies SCH and the concentration has a gamma distribution. The breaking of SCH and the deviation of the concentration distribution from gamma were analyzed in detail in the present paper.

Our model differs from the above models in numerous ways, by (i) including a wider description of relevant biological processes; (ii) capturing complex distributions and power spectra of concentration fluctuations using a novel non-SCH-based approach; (iii) showing that accurate concentration homeostasis results from a complex tradeoff between dosage compensation, balanced biosynthesis, degradation, and the timing of replication; (iv) showing that concentration homeostasis is typically not strong enough to suppress concentration oscillations due to cell division; (v) showing that the number of modes of mRNA number

distributions is greater than or equal to that of mRNA concentration distributions; (vi) showing that a sizer-timer or adder-timer compound strategy is ideal because it maintains accurate concentration homeostasis whilst minimising concentration noise. In the study concentration homeostasis, we disentangled the extrinsic noise due to cell cycle effects from the total concentration variability, and then use their ratio to characterize the accuracy of homeostasis. Our definition of homeostasis can be identified experimentally by the scatter plot of molecule number and cell volume—accurate homeostasis is indicated by a strong linear relationship between mean molecule number and cell volume, while inaccurate homeostasis is indicated by a strong nonlinear relationship between the two.

Our model is complex due to the coupling between gene expression dynamics, cell volume dynamics, and cell cycle events. A natural question is whether all the parameters involved in the model can be inferred accurately. In fact, parameter inference for similar models has been made in our previous papers—the parameters related to cell volume dynamics have been inferred in *E. coli* [52] and fission yeast [53] using the method of distribution matching, and the parameters related to gene expression dynamics have be estimated in *E. coli* [40] using the method of power spectrum matching. In Section G in S1 Appendix, we provide details of a parameter inference method for our model and validate it using synthetic cell-cycle resolved lineage data of gene expression and cell volume obtained from stochastic simulations. We show that fitting this data to the model leads to accurate estimation of all model parameters (see Table A in S1 Appendix for the comparison between input and estimated parameters).

Although our model is detailed, it has some limitations. We briefly discuss a few of the most relevant ones from a physiological point of view: (i) we have assumed no particular change in transcription during mitosis. However, during mitosis, chromosomes are highly condensed, physically excluding the transcription machinery and leading to transcriptional silence; subsequently at mitotic exit, the transcriptional program is faithfully reactivated, allowing the cell to maintain its identity and continue to function [97]; (ii) we have assumed that the parameters $\beta$ and $\kappa$ are independent of each other, while in reality the two are likely related because both balanced biosynthesis and dosage compensation emerge from RNA polymerase dynamics; (iii) we have assumed the growth rate is constant through the cell cycle, while in real systems the growth rate has small fluctuations [45, 49, 98]; (iv) while we assumed that only the synthesis rate is volume dependent, in some cell types both the synthesis and degradation rates of mRNA may be volume-dependent—this may be needed to maintain concentration homeostasis because the synthesis rate does not always scale linearly with cell size [99]; (v) our model does not provide a detailed description of the biochemical processes at play behind the maintenance of concentration homeostasis, such as competition between genes for limiting RNA polymerase II [16] and the negative regulation of RNA polymerase II activity and abundance by nuclear mRNA [20, 91]; (vi) we have focused on the expression of unregulated genes but it is well known that many regulate each other resulting in complex gene regulatory networks [100]. In particular, regarding the latter point, it remains to be seen how the plethora of results on noise-induced bifurcations and oscillations derived using classical models are modified when size-dependent transcription and cell-size control strategies are introduced.

## Methods

### Distributions of the generalized added volumes

Let $V_b$, $V_r$, and $V_d$ denote the cell volumes at birth, replication, and division, respectively. Let $\alpha_0$ and $\alpha_1$ denote the strengths of size control before and after replication, respectively. Here we will prove that (i) the generalized added size before replication, $\Delta_0 = V_r^{\alpha_0} - V_b^{\alpha_0}$, has an Erlang distribution with shape parameter $N_0$ and mean $M_0 = N_0 \alpha_0 g / a^{\alpha_0}$ and (ii) the

generalized added size after replication, $\Delta_1 = V_d^{\alpha_1} - V_r^{\alpha_1}$, is also Erlang distributed with shape parameter $N_1 = N - N_0$ and mean $M_1 = N_1 \alpha_1 g / a^{\alpha_1}$.

To prove this, we first focus on the volume dynamics before replication. Note that when $V_b$ is fixed, the cell volume at time $t$ is given by $V(t) = V_b e^{gt}$. Since the transition rate from one stage to the next is equal to $(aV(t))^{\alpha_0}$ before replication, the distribution of the transition time $T$ is given by

$$\mathbb{P}(T > t) = e^{-\int_0^t (aV(s))^{\alpha_0}\,ds} = e^{-\int_0^t (aV_b)^{\alpha_0} e^{\alpha_0 g s}\,ds} = e^{-\frac{(aV_b)^{\alpha_0}}{\alpha_0 g}(e^{\alpha_0 g t}-1)}.$$

This shows that

$$\mathbb{P}\left(V_b^{\alpha_0}\left(e^{\alpha_0 g T} - 1\right) > t\right) = e^{-\frac{a^{\alpha_0} t}{\alpha_0 g}}. \tag{11}$$

Let $v_k$ be the volume when the cell first enters stage $k$ with $v_1 = V_b$ and $v_{N_0+1} = V_r$. Note that $V_b^{\alpha_0}(e^{\alpha_0 g T} - 1) = v_2^{\alpha_0} - v_1^{\alpha_0}$ is the generalized added size at stage 1. Hence Eq (11) suggests that $v_2^{\alpha_0} - v_1^{\alpha_0}$ is exponentially distributed with mean $\alpha_0 g / a^{\alpha_0}$. Similarly, we can prove that the generalized added size $v_{k+1}^{\alpha_0} - v_k^{\alpha_0}$ at stage $k$ is also exponentially distributed with mean $\alpha_0 g / a^{\alpha_0}$ for each $k \in [1, N_0]$. As a result, the generalized added size before replication, $V_r^{\alpha_0} - V_b^{\alpha_0}$, is the independent sum of $N_0$ exponentially distributed random variables with the same mean $\alpha_0 g / a^{\alpha_0}$. This indicates that $V_r^{\alpha_0} - V_b^{\alpha_0}$ has an Erlang distribution with shape parameter $N_0$ and mean $M_0 = N_0 \alpha_0 g / a^{\alpha_0}$, and is also independent of $V_b$. The fact that $V_d^{\alpha_1} - V_r^{\alpha_1}$ is Erlang distributed with shape parameter $N_1$ and mean $M_1$, and is also independent of both $V_b$ and $V_r$ can be proved in the same way.

Moreover, it is easy to see that for each $k \in [1, N_0]$, the generalized added size $v_{k+1}^{\alpha_0} - v_1^{\alpha_0}$, has an Erlang distribution with shape parameter $k$ and mean $k\alpha_0 g / a^{\alpha_0}$. Similarly, for each $k \in [N_0 + 1, N]$, the generalized added size $v_{k+1}^{\alpha_1} - v_{N_0+1}^{\alpha_1}$ is also Erlang distributed with shape parameter $k - N_0$ and mean $(k - N_0)\alpha_1 g / a^{\alpha_1}$.

## Effective size control strategy over the whole cell cycle

In our model, we assume that the size control strategies before and after replication can be different with the strength of size control changing from $\alpha_0$ to $\alpha_1$ at replication. The following question naturally arises: what is the size control strategy over the whole cell cycle? To answer this, we first focus on the cell size dynamics before replication. Recall that the generalized added size $V_r^{\alpha_0} - V_b^{\alpha_0}$ before replication has an Erlang distribution and is independent of $V_b$. When the fluctuations of $V_b$ and $V_r$ are small, we have the following Taylor expansions:

$$V_b^{\alpha_0} \approx v_b^{\alpha_0} + \alpha_0 v_b^{\alpha_0-1}(V_b - v_b) = (1 - \alpha_0)v_b^{\alpha_0} + \alpha_0 v_b^{\alpha_0-1} V_b,$$
$$V_r^{\alpha_0} \approx v_r^{\alpha_0} + \alpha_0 v_r^{\alpha_0-1}(V_r - v_r) = (1 - \alpha_0)v_r^{\alpha_0} + \alpha_0 v_r^{\alpha_0-1} V_r,$$

where $v_b = v_1$ and $v_r = v_{N_0+1}$ are the typical values of $V_b$ and $V_r$, respectively. Therefore, we obtain

$$\begin{aligned}
V_r^{\alpha_0} - V_b^{\alpha_0} &\approx (1 - \alpha_0)(v_r^{\alpha_0} - v_b^{\alpha_0}) + \alpha_0 v_r^{\alpha_0-1} V_r - \alpha_0 v_r^{\alpha_0-1} V_b \\
&= (1 - \alpha_0)(v_r^{\alpha_0} - v_b^{\alpha_0}) + \alpha_0 v_r^{\alpha_0-1}\left[V_r - \left(\frac{v_b}{v_r}\right)^{\alpha_0-1} V_b\right] \\
&= (1 - \alpha_0)(v_r^{\alpha_0} - v_b^{\alpha_0}) + \alpha_0 v_r^{\alpha_0-1}(V_r - 2^{(1-\alpha_0)w} V_b),
\end{aligned}$$

where $w = \log_2(v_r/v_b)$ is the proportion of cell cycle before replication. This shows that

$$V_r^{\alpha_0} - V_b^{\alpha_0} \approx a + b(V_r - 2^{(1-\alpha_0)w} V_b),$$

where $a = (1 - \alpha_0)(v_r^{\alpha_0} - v_b^{\alpha_0})$ and $b = \alpha_0 v_r^{\alpha_0 - 1}$ are two constants. Hence $V_r - 2^{(1-\alpha_0)w} V_b$ is approximately independent of $V_b$, i.e.

$$V_r - 2^{w(1-\alpha_0)} V_b = \epsilon_0, \tag{12}$$

where $\epsilon_0$ is a noise term independent of $V_b$ and which has a mean greater than zero. Similarly, for the cell size dynamics after replication, we have

$$V_d - 2^{(1-w)(1-\alpha_1)} V_r = \epsilon_1, \tag{13}$$

where $\epsilon_1$ is a noise term independent of both $V_b$ and $V_r$, and which has a mean greater than zero. Combining the above two equations, we can eliminate $V_r$ and obtain

$$V_d - 2^{1-[w\alpha_0 + (1-w)(1-\alpha_1)]} V_b = \epsilon_1 + 2^{(1-w)(1-\alpha_1)} \epsilon_0, \tag{14}$$

where the right-hand side is a noise term independent of $V_b$. Eqs (12) and (13) again show that $\alpha_i \to 0, 1, \infty$ corresponds to timer, adder, and sizer, respectively.

To proceed, note that if the strength of size control does not change upon replication, i.e. $\alpha_0 = \alpha_1 = \alpha$, then similar reasoning suggests that $V_d - 2^{1-\alpha} V_b$ is a noise term independent of $V_b$. Therefore, if the strength of size control changes from $\alpha_0$ to $\alpha_1$ at replication, then it follows from Eq (14) that the effective size control strength $\alpha$ over the whole cell cycle should be defined as

$$\alpha = w\alpha_0 + (1 - w)\alpha_1.$$

The cell behaves as an overall timer/adder/sizer when $\alpha \to 0, 1, \infty$, respectively.

## Lineage distributions of gene product number

We consider the distribution of copy numbers for lineage measurements. Note that at each cell cycle stage, the stochastic dynamics of copy number is exactly the classical discrete bursty model proposed in [88], whose steady-state is the negative binomial distribution. Therefore, it is natural to approximate the conditional distribution of copy numbers at each stage by the negative binomial distribution

$$p_{n|k} \approx \frac{1}{n!} \left( \frac{\bar{\mu}_k^2}{\bar{\sigma}_k^2 - \bar{\mu}_k} \right)_n \left( 1 - \frac{\bar{\mu}_k}{\bar{\sigma}_k^2} \right)^n \left( \frac{\bar{\mu}_k}{\bar{\sigma}_k^2} \right)^{\bar{\mu}_k^2/(\bar{\sigma}_k^2 - \bar{\mu}_k)},$$

where $\bar{\mu}_k = m_{1k}/m_{0k}$ and $\bar{\sigma}_k^2 = (m_{1k} + m_{2k})/m_{0k} - \bar{\mu}_k^2$ are the conditional mean and variance of copy number at stage $k$, respectively, and $(x)_n = x(x+1)\cdots(x+n-1)$ denotes the Pochhammer symbol. Hence the overall distribution of copy numbers is given by the following mixture of $N$ negative binomial distributions:

$$p_n = \sum_{k=1}^{N} m_{0k} p_{n|k} \approx \sum_{k=1}^{N} \frac{m_{0k}}{n!} \left( \frac{\bar{\mu}_k^2}{\bar{\sigma}_k^2 - \bar{\mu}_k} \right)_n \left( 1 - \frac{\bar{\mu}_k}{\bar{\sigma}_k^2} \right)^n \left( \frac{\bar{\mu}_k}{\bar{\sigma}_k^2} \right)^{\bar{\mu}_k^2/(\bar{\sigma}_k^2 - \bar{\mu}_k)}. \tag{15}$$

We emphasize that the above derivation is not rigourous; the analytical distribution derived is not exact but it serves an accurate approximation when $N \geq 15$. The logic behind our approximate distribution is that while the gene product may produce complex distribution of copy numbers, when the number of cell cycle stages is large, the distribution must be relatively

simple within each stage and thus can be well approximated by a simple negative binomial distribution.

## Lineage distributions of gene product concentration

We consider the distribution of concentration for lineage measurements. Note that at each cell cycle stage, the stochastic dynamics of concentration is exactly the classical continuous bursty model proposed in [87], whose steady-state is the gamma distribution. Therefore, it is natural to approximate the conditional distribution of concentrations at each stage by the gamma distribution

$$p(x|k) \approx \frac{(\mu_k/\sigma_k^2)^{\mu_k^2/\sigma_k^2}}{\Gamma(\mu_k^2/\sigma_k^2)} x^{\mu_k^2/\sigma_k^2 - 1} e^{-\mu_k x/\sigma_k^2},$$

where $\mu_k = m_{1k}/(m_{0k}v_k)$ and $\sigma_k^2 = (m_{1k} + m_{2k})/(m_{0k}v_k^2) - \mu_k^2$ are the conditional mean and variance of concentration at stage $k$, respectively. Hence the overall distribution of concentrations is given by the following mixture of $N$ gamma distributions:

$$p(x) = \sum_{k=1}^{N} m_{0k} p(x|k) \approx \sum_{k=1}^{N} \frac{m_{0k}(\mu_k/\sigma_k^2)^{\mu_k^2/\sigma_k^2}}{\Gamma(\mu_k^2/\sigma_k^2)} x^{\mu_k^2/\sigma_k^2 - 1} e^{-\mu_k x/\sigma_k^2}.$$

We emphasize that the above derivation is not rigourous; the analytical distribution derived is not exact but it serves an accurate approximation when $N \geq 15$. While the gene product may produce complex distribution of concentrations, when $N$ is sufficiently large, the distribution must be relatively simple within each stage and thus can be well approximated by a simple gamma distribution.

## Population distributions of gene product number and concentration

Up till now and in the main text, we have calculated gene product statistics for lineage measurements. Here we perform calculations for population measurements following the approach described in [37, 38]. Let $C_k(t)$ be the mean number of cells at stage $k$ in the population at time $t$. Then all $C_k(t)$, $1 \leq r \leq N$ satisfy the following set of differential equations:

$$\dot{C}_1 = -q_1 C_1 + 2q_N C_N,$$

$$\dot{C}_2 = -q_2 C_2 + q_1 C_1,$$

$$\cdots \qquad\qquad\qquad\qquad (16)$$

$$\dot{C}_N = -q_N C_N + q_{N-1} C_{N-1},$$

where the term $2q_N C_N$ in the first equation is due to the fact that the cell divides when the system transitions from stage $N$ to stage 1. We then make the ansatz

$$C_k(t) = \pi_k C(t),$$

where $C(t)$ is the total number of cells in the population and $\pi_k$ is the proportion of cells at

stage $k$. Then Eq (16) can be rewritten as

$$\pi_1 \dot{C} = (2q_N \pi_N - q_1 \pi_1)C,$$

$$\pi_2 \dot{C} = (q_1 \pi_1 - q_2 \pi_2)C,$$

$$\cdots$$

$$\pi_N \dot{C} = (q_{N-1} \pi_{N-1} - q_N \pi_N)C.$$

It then follows from the above equation that

$$2q_N \frac{\pi_N}{\pi_1} - q_1 = q_1 \frac{\pi_1}{\pi_2} - q_2 = \cdots = q_{N-1} \frac{\pi_{N-1}}{\pi_N} - q_N = J,$$

where $J = \dot{C}/C$. This clearly shows that

$$\frac{\pi_N}{\pi_1} = \frac{q_1 + J}{2q_N}, \quad \frac{\pi_1}{\pi_2} = \frac{q_2 + J}{q_1}, \quad \cdots, \frac{\pi_{N-1}}{\pi_N} = \frac{q_N + J}{q_{N-1}}. \tag{17}$$

Multiplying all the above equations shows that $J$ is the solution to the equation

$$(q_1 + J)(q_2 + J) \cdots (q_N + J) = 2q_1 q_2 \cdots q_N.$$

It thus follows from Eq (17) that

$$\pi_k = \frac{(q_{k+1} + J) \cdots (q_N + J)}{q_k \cdots q_{N-1}} \pi_N.$$

Since all $\pi_k$ add up to 1, we finally obtain

$$\pi_k = \frac{(q_{k+1} + J) \cdots (q_N + J)/(q_k \cdots q_{N-1})}{\sum_{k=1}^{N} (q_{k+1} + J) \cdots (q_N + J)/(q_k \cdots q_{N-1})}.$$

Now we have obtained the proportion of cells at each stage. Therefore, the population distribution of copy number is given by the following mixture of $N$ negative binomial distributions:

$$p_n^{\text{pop}} = \sum_{k=1}^{N} \pi_k p_{n|k} \approx \sum_{k=1}^{N} \frac{\pi_k}{n!} \left( \frac{\bar{\mu}_k^2}{\bar{\sigma}_k^2 - \bar{\mu}_k} \right)_n \left( 1 - \frac{\bar{\mu}_k}{\bar{\sigma}_k^2} \right)^n \left( \frac{\bar{\mu}_k}{\bar{\sigma}_k^2} \right)^{\bar{\mu}_k^2/(\bar{\sigma}_k^2 - \bar{\mu}_k)}.$$

Similarly, the population distribution of concentration is given by the following mixture of $N$ gamma distributions:

$$p_{\text{pop}}(x) = \sum_{k=1}^{N} \pi_k p(x|k) \approx \frac{\pi_k (\mu_k/\sigma_k^2)^{\mu_k^2/\sigma_k^2}}{\Gamma(\mu_k^2/\sigma_k^2)} x^{\mu_k^2/\sigma_k^2 - 1} e^{-\mu_k x/\sigma_k^2}.$$

## Supporting information

**S1 Appendix. Technical details.** This file contains the detailed derivation of some equations, the technical details of some figures, and the detailed description of the parameter inference method.
(PDF)

**S1 Fig. Comparison between the mRNA distributions the bursty model and the three-stage model with a cell volume description.** The blue (red) dots show the simulated mRNA concentration (copy number) distribution for the three-stage model with a cell volume description obtained from SSA. The blue (red) curves show the simulated mRNA concentration (copy number) distribution for the bursty model with a cell volume description obtained from SSA. The parameters for the three-stage model are chosen as $N_0 = 0.2N$, $\beta = 1$, $\kappa = 2$, $\alpha_0 = \alpha_1 = 1$, $v = 10f$, $d = f$, $s = 0.2r$, $u = v$. The parameters for the bursty model are chosen as $N_0 = 0.2N$, $B = 0.2$, $\beta = 1$, $\kappa = 2$, $\alpha_0 = \alpha_1 = 1$, $v = 10f$, where $v$ is the degradation rate of mRNA. The growth rate $g$ is determined so that $f = 2$ for the bursty model. The parameter $\rho$ and $a$ are chosen so that the mean mRNA number $\langle n \rangle = 25$ and the mean cell volume $\langle V \rangle = 1$ for the bursty model.
(EPS)

**S2 Fig. Comparison between the protein distributions for bursty model and the three-stage model with a cell volume description.** The blue (red) dots show the simulated protein concentration (copy number) distribution for the three-stage model with a cell volume description obtained from SSA. The blue (red) curves show the simulated protein concentration (copy number) distribution for the bursty model with a cell volume description obtained from SSA. The parameters for the three-stage model are chosen as $N_0 = 0.2N$, $\rho = 70$, $\beta = 1$, $\kappa = 2$, $\alpha_0 = \alpha_1 = 1$, $d = f$, $r = 50\rho$, $s = 0.2r$, $u = v$. The parameters for the bursty model are chosen as $N_0 = 0.2N$, $\rho = 70$, $B = 0.2$, $\beta = 1$, $\kappa = 2$, $\alpha_0 = \alpha_1 = 1$, $d = f$, where $d$ is the degradation rate of protein. The growth rate $g$ is determined so that $f = 2$ for the bursty model. The parameter $a$ are chosen so that the mean cell volume $\langle V \rangle = 1$ for the bursty model.
(EPS)

**S3 Fig. Comparison between the full and mean-field models as $N_0$ and $\eta$ vary.** The blue (red) dots show the simulated concentration (copy number) distribution for the full model obtained from SSA. The blue (red) squares show the simulated concentration (copy number) distribution for the mean-field model obtained from SSA. The blue (red) curve shows the analytical approximate concentration (copy number) distribution. The model parameters are chosen as $N = 50$, $B = 1$, $\beta = 0$, $\kappa = 2$, $\alpha_0 = \alpha_1 = 1$, $d = \eta f$. The growth rate $g$ is determined so that $f = 0.1$. The parameter $\rho$ and $a$ are chosen so that $\langle n \rangle = 100$ and $\langle V \rangle = 1$.
(EPS)

**S4 Fig. Power spectra of copy number and concentration fluctuations. A**: Power spectra of copy number fluctuations when $w = 0.5$ (blue curve) and $w = 0.8$ (red curve). **B**: Power spectra of concentration fluctuations when $w = 0.5$ (blue curve) and $w = 0.8$ (red curve). In A,B, the model parameters are chosen as $N = 20$, $B = 1$, $\beta = 0$, $\kappa = \sqrt{2}$, $\alpha_0 = \alpha_1 = 0.01$, $d = 0.5$, $\eta = 5$. The growth rate $g$ is determined so that $f = 0.1$. The parameters $\rho$ and $a$ are chosen so that $\langle n \rangle = 100$ and $\langle V \rangle = 1$. The power spectra are computed using Eq (17) in S1 Appendix.
(EPS)

**S5 Fig. Relationship between concentration homeostasis and concentration oscillations. A**: Heat plot of $\gamma$ (the accuracy of concentration homeostasis) versus $B$ (the mean burst size) and $\langle n \rangle$ (the mean gene product number). **B**: Heat plot of $H$ (the height of the off-zero peak of the power spectrum) versus $B$ and $\langle n \rangle$. The model patenters in A,B are chosen as $N = 40$, $N_0 = 0.4N$, $\beta = 1$, $\kappa = 2$, $\alpha_0 = \alpha_1 = 1$, $d = 0.1$, $\eta = 1$. The growth rate $g$ is determined so that $f = 0.1$. The parameter $a$ is chosen so that $\langle V \rangle = 1$.
(EPS)

**S6 Fig. Influence of two-layer cell-size control on concentration homeostasis and concentration noise for various choices of $\beta$ and $\kappa$. A-C**: Heat plot of the homeostasis accuracy $\gamma$ (left panel) and concentration noise $\phi$ (right panel) versus the strengths of size control, $\alpha_0$ and $\alpha_1$. The model parameters are chosen as $N = 50$, $N_0 = 21$, $\rho = 4.7$, $B = 1$, $d = 0.4$, $\eta = 4$.
(EPS)

**S7 Fig. Cell volume distribution in budding yeast.** The blue curve shows the volume distribution of all cells. The green curve shows the distribution of cells in the $G_1$ phase and the yellow curve shows the distribution of cells in the $G_2$/S/M phase. The data shown are published in [92].
(EPS)

**S8 Fig. Copy number and concentration distributions of three mRNAs in budding yeast. A**: The mRNA copy number (red curve) and concentration (blue curve) distributions for the *HTB1* gene. The green curves show the distributions of cells in the $G_1$ phase and the yellow curves show the distributions of cells in the $G_2$/S/M phase. **B**: Same as A but for the *ACT1* gene. **C**: Same as A but for the *HHF1* gene. The data shown are published in [92].
(EPS)

## Author Contributions

**Conceptualization:** Chen Jia, Abhyudai Singh, Ramon Grima.

**Investigation:** Chen Jia.

**Methodology:** Chen Jia.

**Project administration:** Ramon Grima.

**Supervision:** Abhyudai Singh, Ramon Grima.

**Writing – original draft:** Chen Jia, Ramon Grima.

**Writing – review & editing:** Chen Jia, Abhyudai Singh, Ramon Grima.

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
