## [Decision Letter · Decision Letter 0]

11 Aug 2022

Dear Prof. Grima,

Thank you very much for submitting your manuscript "Concentration fluctuations due to size-dependent gene expression and cell-size control mechanisms" for consideration at PLOS Computational Biology. As with all papers reviewed by the journal, your manuscript was reviewed by members of the editorial board and by several independent reviewers. The reviewers appreciated the attention to an important topic. Based on the reviews, we are likely to accept this manuscript for publication, providing that you modify the manuscript according to the review recommendations. In particular, following Reviewer 1, we would ask the Authors to "motivate this G(0)=1 normalization and explain more intuitively what C(β,κ,ω) really quantifies." They should also discuss, as suggested by Reviewer 2, the generality of the threshold accumulation processes as opposed to dilution of inhibitors (es Whi5 in budding yeast), and how that could affect their model.

Sincerely,

Andrea Ciliberto

Associate Editor

PLOS Computational Biology

Jason Haugh

Deputy Editor

PLOS Computational Biology

[LINK]

Reviewer's Responses to Questions

**Comments to the Authors:**

Reviewer #1: In this revised version of the manuscript by Jia et al., the authors have appropriately addressed all of my comments, except one, for which clarifications or modifications would further improve the manuscript.

This is regarding the earlier comment I made on the power spectrum analysis in my first review. This comment was motivated by the same idea as the one just before, i.e. that the metric for quantifying homeostasis should be a dimensionless measure of the relative contribution of oscillations to the total fluctuations (which includes both a cell-cycle-driven component and a stochastic-bursting-driven component). If the metric you use is the height of the peak, it should be normalized to something that somehow measures the total fluctuations. I understand now that G(0) is normalized to 1 even in the calculations (I thought it was just on the plots) and realize that C(β,κ,ω) is already dimensionless. So you might already be doing what I have in mind. But in that case, you should motivate this G(0)=1 normalization and explain more intuitively what C(β,κ,ω) really quantifies. You did it very well with γ (explaining that φ has two components and that γ quantifies the relative contribution of the extrinsic oscillations φext to the total φ). Something similar for the power spectrum would be a big plus.

The manuscript has improved in clarity since the first submission. I think it is elegant and valuable work, which will benefit to the community.

Reviewer #2: Jia and coworkers propose a model of stochastic gene expression including several effects that are typically disregarded, such as cell-cycle effects, dosage and cell-division control. I have been asked to review this work with the explicit question of evaluating to what extent the issues raised by reviewer three would prevent acceptance.

My impression is that the work is suitable for publication on PLOS computational biology, as most if not all of the concerns raised by this reviewer have to do with modeling choices and assumptions. Clearly, different choices are possible, and they have consequences, but the current state of experimental knowledge on all the processes considered in this model is insufficient to fully determine these choices.

Probably this is not the ultimate model of this kind, as several choices and assumptions will need the support of experimental data. However, the authors present a number of interesting testable predictions, and discuss how they could relate to existing data.

However, I would encourage the authors to perform an additional round of writing where they clearly separate assumptions from their justifications, and from validation attempts with published results or experimental data. In particular, this refers to the points 6.7 and 14 raised by reviewer 3. I also note that the processes considered in this study are not the sole ones that may play a role in concentration homeostasis, and this may emerge e.g. in the discussion.

As specific example, I find that some of the statements made by the authors to justify the model (also in reply to the reviewers) are fragile. For example, it is not true that in E. coli it is established that a threshold accumulation process related to FtsZ sets cell division, and almost surely things are more complex than that. Equally, for yeast, things are complex and not well understood, and surely inhibitor dilution processes were argued to be very important at least for some cell cycle stages.

Additionally, for the question of the model of (single-gene) replication timing I agree with the referee that the question is complex (the authors may want to check PMID for a model of stochastic replication timing in yeast, and PMID: 34795646 - and references therein - for E coli).

This said, I do not think that any of the above comments should prevent acceptance, it is just a matter of properly calibrating the authors' claims and presentation of the model. In these respects, the comments from reviewer 3 are valid. However, I think this does not undermine the value of the model, as long as the assumptions are clearly presented as assumptions, and possible alternatives, limitations and caveats are discussed honestly.

**Have the authors made all data and (if applicable) computational code underlying the findings in their manuscript fully available?**

Reviewer #1: Yes

Reviewer #2: Yes

PLOS authors have the option to publish the peer review history of their article (what does this mean?). If published, this will include your full peer review and any attached files.

Reviewer #1: No

Reviewer #2: No

Figure Files:

Data Requirements:

Reproducibility:

References:

---

## [Editor Report · Decision Letter 1]

14 Sep 2022

Dear Prof. Grima,

We are pleased to inform you that your manuscript 'Concentration fluctuations in growing and dividing cells: insights into the emergence of concentration homeostasis' has been provisionally accepted for publication in PLOS Computational Biology.

Best regards,

Andrea Ciliberto

Academic Editor

PLOS Computational Biology

Jason Haugh

Section Editor

PLOS Computational Biology

---

## [Editor Report · Acceptance letter]

28 Sep 2022

PCOMPBIOL-D-22-01042R1 

Concentration fluctuations in growing and dividing cells: insights into the emergence of concentration homeostasis

Dear Dr Grima,

I am pleased to inform you that your manuscript has been formally accepted for publication in PLOS Computational Biology. Your manuscript is now with our production department and you will be notified of the publication date in due course.

With kind regards,

Anita Estes
